# Multi-Label Node Classification with Label Influence Propagation

**Yifei Sun**[1*], **Zemin Liu**[1*], **Bryan Hooi**[2], **Yang Yang**[1*], **Rizal Fathony**[3], **Jia Chen**[4], **Bingsheng He**[2*]
[1]Zhejiang University, [2]National University of Singapore,
[3]Capital One, [4]GrabTaxi Holdings Pte. Ltd.
{yifeisun, liu.zemin, yangya}@zju.edu.cn,
{bhooi, hebs}@comp.nus.edu.sg,
rizal.fathony@capitalone.com, jia.chen@grab.com

## Abstract

Graphs are a complex and versatile data structure used across various domains, with possibly multi-label nodes playing a particularly crucial role. Examples include proteins in PPI networks with multiple functions and users in social or e-commerce networks exhibiting diverse interests. Tackling multi-label node classification (MLNC) on graphs has led to the development of various approaches. Some methods leverage graph neural networks (GNNs) to exploit label co-occurrence correlations, while others incorporate label embeddings to capture label proximity. However, these approaches fail to account for the intricate influences between labels in non-Euclidean graph data. To address this issue, we decompose the message passing process in GNNs into two operations: *propagation* and *transformation*. We then conduct a comprehensive analysis and quantification of the influence correlations between labels in each operation. Building on these insights, we propose a novel model, Label Influence Propagation (**LIP**). Specifically, we construct a label influence graph based on the integrated label correlations. Then, we propagate high-order influences through this graph, dynamically adjusting the learning process by amplifying labels with positive contributions and mitigating those with negative influence. Finally, our framework is evaluated on comprehensive benchmark datasets, consistently outperforming SOTA methods across various settings, demonstrating its effectiveness on MLNC tasks[1].

## 1 Introduction

Graphs, as a complex data structure, are prevalent across various fields (Jiang et al., 2019; Kipf & Welling, 2016; Ying et al., 2018; Liu et al., 2023; Fang et al., 2024). Among these, graphs with multi-label nodes are common and of great importance. For instance, proteins in ogbn-protein dataset have multiple functions (Hu et al., 2020). Accurately identifying all the functions can assist with understanding biological processes and advancing biomedical research. Thus, we focus on this realistic but challenging problem named multi-label node classification on graphs, which we abbreviate as MLNC in the following paper.

**Prior studies.** Current methods typically adopt three strategies to address this problem. The *first strategy* is to neglect the multi-label information and predict the labels without mining label correlations (Shi et al., 2020b; Li et al., 2023). The *second strategy* is to explicitly treat labels as a new type of node and incorporate them into the original graph, thereby enhancing task performance through propagation and aggregation information between nodes and label nodes (Gao et al., 2019; Shi et al., 2020a). Since only incomplete connections between nodes and label nodes are available in the training set, the *third strategy* is to integrate label representations into the neighbor aggregation and classification processes, thereby improving the utilization of multi-label information (Zhou et al., 2021; Xiao et al., 2022). However, these strategies underestimate the complex label correla-

---

[*]Corresponding authors.
[1]Our code is available at `https://github.com/Xtra-Computing/LIP_MLNC`.

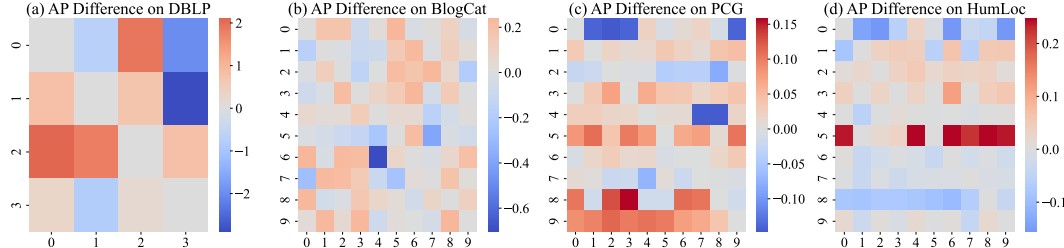

Figure 1: Observations showing positive (red) and negative (blue) influence between different labels. Each value in the heat maps[3] represents the performance on the column label when trained together with the row label, minus the result of training the column label individually.

tions in graph data, only assessing label proximity without modeling their influence, thereby failing to fully utilize these correlations to improve MLNC.

**Observations.** Our observations (Fig. 1) further show that, for graph data, different labels can mutually enhance or harm each other's performance. Specifically, we compare the performance differences when training a graph neural network (GNN) using both row and column labels simultaneously versus training with only column labels individually. A positive difference indicates that jointly training with row and column labels outperforms training with column label alone, implying that the row label can enhance the column label. Conversely, a negative difference suggests a negative influence. As shown in Fig. 1, most graph datasets exhibit both positive and negative influence between labels. Therefore, in this paper we aim to analyze and quantify these complex label influences on graphs to enhance or mitigate the positive or negative effects.

**Challenges.** However, given the intricate nature of graph, several challenges emerges. First, the influences existing not only between labels themselves but between nodes entangled together through graph structure. Thus, *how to quantify the influences between labels on the complex non-Euclidean graph data* is the first challenge. Second, since any label may exert both positive and negative influences on multiple other labels simultaneously, the second challenge is *how to capture the high-order influence correlations between all labels*. Finally, our goal is to encourage or suppress labels that bring positive or negative influences, respectively. Thus, *how to leverage the quantified high-order influence correlations to enhance the performance of MLNC task* is the third challenge.

**Present work.** To address the *first* challenge, we decompose the message passing into propagation and transformation operations (Zhang et al., 2022a), allowing for a detailed analysis of the label influence correlation on graph data. Moreover, we provide a theoretical analysis, grounded in the inductive biases inherent in graph models (Xu et al., 2018a; Wang & Leskovec, 2020), revealing that the graph structure itself is a key driver of label influence during propagation operation. As for transformation operation, our analysis revealed that the dynamic interactions between labels during training are caused during each back propagation on model parameters. The above two parts of the label influence analysis not only provide a deep insight into multi-label correlations on graph data but also lay a solid foundation for improving MLNC on graphs.

To tackle the *second* challenge, we construct a high-order label influence propagation graph based on the pair-wise correlations. Thus, we can quantify the propagated influence from one label to any other labels on graph data. For the *third* challenge, we propose a novel method, Label Information Propagation (**LIP**), which contains the above quantification steps and crafts the training dynamics among multiple labels to improve the overall performance. By calculating the importance of each label in this propagation graph, we dynamically adjusted its learning proportion throughout the training process. This ensures that labels with more positive influence on others are learned more effectively, while minimizing negative interactions between labels, ultimately enhancing the performance of MLNC. As a plug-and-play approach, our method can be applied to various GNN backbones. We validated its effectiveness using multiple settings on the most comprehensive collection (Zhao et al., 2023) of MLNC datasets.

---

[3]These heat maps are asymmetric because each value represents the degree to which the row label helps ($+$) or harms ($-$) the column label. Without loss of generality, we selected the first 10 labels as observation targets (except for DBLP) due to space limitations.

**Contributions.** To summarize, our contributions are three-fold.

- We offer a novel perspective on label correlation analysis by dissecting the pipeline into forward and backward propagation segments, where we conduct a thorough analysis and quantification of the influence correlation between labels.

- Building on the quantified pairwise label influence, we design **LIP**, which calculates high-order label correlation and the influence propagation between labels to dynamically guide the learning process of the model.

- Empirically, we validate the superiority of our method on MLNC graph data across various domains, showing improvements (AUC) of 3.06% and 3.42% on average under the node split and label split settings, respectively.

## 2 RELATED WORK

**Multi-label Node Classification (MLNC).** To promote research on MLNC, MLGNC (Zhao et al., 2023) provides three biological datasets and performs a detailed analysis on both datasets and existing methods. From the methodology perspective, some methods consider using textual embeddings of labels to incorporate label proximity. LARN (Xiao et al., 2022) incorporates the all label embeddings during the neighbor aggregation process. Similarly, LANC (Zhou et al., 2021) proposes attention mechanism to aggregates the label embeddings with each node. Another line of work construct additional label-label network to use the label co-occurrence correlations. ML-GCN (Gao et al., 2019) construct both node-centric and label-centric graphs to aggregate information, while MLNE (Shi et al., 2020a) use random walk to get embedding on both label-label network and original graph. There are also some methods that modify the approach to modeling graphs from multi-label perspective. MLGW (Akujuobi et al., 2019) leverage label-specific agents to walk on graphs. VariMul (Song et al., 2021) utilizes VGAE to derive node and label embeddings and designs a confidence ranking loss to mine pairwise label correlations. Recent work improves MLNC by addressing the ambiguity in multi-label graph structures (Bei et al., 2024) and the limited expressive power for multi-label tasks (Zhao & Khosla, 2024). However, these work do not explicitly analyze and quantify the influences between labels and exploit them to enhance the MLNC task.

**Decoupled GNNs.** Compared to conventional GNNs, decoupled GNNs explicitly isolate the two operations and aggregate all propagation (P) and transformation (T) operations separately for processing. APPNP Gasteiger et al. (2019) and DAGNN Liu et al. (2020) first propose that decoupling the two operations can enable GNNs to go much deeper without causing over-smoothing. Although SGC Wu et al. (2019) and lightGCN He et al. (2020) do not discuss the decouple paradigm, it demonstrate superior efficiency and performance. PTA Dong et al. (2021) further explore the paradigm from the perspective of label propagation. Since the APPNP is more suitable for homophily graphs, GPRGNN Chien et al. (2020) improves it by adaptively learning GPR weights. This method not only avoids over-smoothing but also enables adaptive filtering to handle heterophily graphs. NCGNN Chen et al. (2024a) improves model capacity and training efficiency by pre-propagating node features, then employs a CNN-based approach to aggregate the propagated features, adaptively capturing contextual information to boost the performance. Our work is inspired by above studies that decouple or disentangle Zhang et al. (2022a) the message passing process. We focus more on analyzing the influence correlations between labels during the P and T operations, rather than designing specific P and T operations to increase model performance Gasteiger et al. (2019); Chien et al. (2020); Sun et al. (2022) or enhance model capacity and scalability Chen et al. (2024a).

## 3 PRELIMINARIES

### 3.1 PROBLEM FORMULATION

Given a graph $G = (V, E)$ with the adjacency matrix $\mathbf{A} \in \mathbb{R}^{n \times n}$, where $n = |V|$ and $e = |E|$ are the total number of nodes and edges. Each node $v_i \in V$ is associated with a feature vector $\mathbf{x}_i \in \mathbb{R}^f$, and all these feature vectors constitute the feature matrix $\mathbf{X} \in \mathbb{R}^{n \times f} = [\mathbf{x}_1, \mathbf{x}_2, \dots, \mathbf{x}_n]^T$ of the graph $G$. Among all the nodes, $n_l$ of $n$ are labeled while the remaining $n_u$ of $n$ are not. For every node $v_i$ in the labeled dataset $D_l = \{(v_i, y_i^{\text{seq}}) \mid 1 \leq i \leq n_l\}$, it also contains a set of labels $y_i^{\text{seq}} \subseteq \mathbf{Y}$ drawn

from the given label space $\mathbf{Y} = \{y_1, y_2, \ldots, y_k\}$ with $k$ possible class labels at most. Furthermore, the label sets for the rest of the unlabeled nodes $D_u = \{v_i \mid n_l + 1 \leq i \leq n_l + n_u\}$ are not available. The goal of MLNC on graphs under the semi-supervised learning setting is to learn a model $h : V \to 2^{\mathbf{Y}}$ with training data $D = D_l \cup D_u$ by taking both graph structure $\mathbf{A}$ and node feature $\mathbf{X}$ into consideration. Moreover, we denote the nodes with label $y_j \in \mathbf{Y}$ ($1 \leq j \leq k$), as a node set $Y_j = \{v_1, v_2, \ldots, v_{m_j}\}$, where $m_j$ is the number of nodes belong to label $y_j$.

## 3.2 Decomposition of MLNC Pipeline

Since MLNC is considered as correlated multiple binary classification problems on graphs, each determining the presence or absence of a certain label (Zhang & Zhou, 2013). If a multi-label task is modeled as unrelated single-label tasks, it not only wastes the information brought by the other labels but also squanders model parameters. Therefore, almost all the multi-label methods adopt a common style which utilize a graph encoder $\Phi_\theta$ as the backbone to explore both the individual information of multiple labels and their correlation information. The graph encoder $\Phi_\theta$ takes in the graph structure $\mathbf{A}$ and the node feature $\mathbf{X}$ and generate node embeddings $\mathbf{Z} \in \mathbb{R}^{n \times d}$ as

$$\mathbf{Z} = \Phi_\theta(\mathbf{A}, \mathbf{X}), \tag{1}$$

where backbone $\Phi_\theta$ parameterized by $\theta$ can be any existing GNNs, such as GCN (Kipf & Welling, 2017), GAT (Velicković et al., 2018), GraphSAGE (Hamilton et al., 2017).

Thereafter, the node embeddings $\mathbf{Z}$ are further mapped into different label spaces by multiple shallow classifiers $\Psi = \{\psi_1, \psi_2, \ldots, \psi_k\}$ with fewer parameters. For each label $y_j$, the objective is to train a classifier $\psi_j$ that computes the prediction $\tilde{\boldsymbol{y}}_j \in \mathbb{R}^n$ as

$$\tilde{\boldsymbol{y}}_j = \psi_j(\mathbf{Z}). \tag{2}$$

By considering both Eqs. 1 and 2 simultaneously, we observe that Eq. 1 is designed to generate node representations that are *shared* across all labels (*i.e.*, all classifiers) for subsequent tasks such as label prediction. In contrast, Eq. 2 is specific to the label $y_j$ and is utilized *privately* by the corresponding classifier $\psi_j$ for label prediction. Consequently, the architecture of the entire framework can be delineated into two main components: the *shared component* and the *private component*, with $\mathbf{Z}$ serving as the central bridge of the two components.

**Definition 1** (Shared component in MLNC). *The shared component in MLNC boils down to the GNN backbone, parameterized by $\theta$, which can be denoted as $\Phi_\theta(\mathbf{A}, \mathbf{X})$. This component is to encode nodes collectively to generate embeddings $\mathbf{Z}$, irrespective of their associated labels.*

**Definition 2** (Private components in MLNC). *The private components in MLNC boils down to the classification head for each label $y_j$, denoted as $\psi_j$, corresponding to the operations $\mathbf{W}_j^{-1}\mathbf{Y}_j$. This component is tasked with calibrating the node embeddings $\mathbf{Z}$ for label-specific prediction.*

## 4 Label Influence Propagation

### 4.1 Overview

The core idea of **LIP** is to capture the influence between labels to guide the training process of model that need to be encouraged or suppressed within the overall set of labels. To achieve this, we first need to quantify the relationships of mutual influence between labels. Then, based on this influence relationship, we further capture high-order relationship between labels and calculate the importance score of loss corresponding to each label during training. Finally, we combine the losses with their respective importance levels to improve the MLNC tasks.

**The analysis from message passing.** While the graph structure makes analyzing label influence relationships highly challenging, these influences arise through the shared GNN backbone decomposed from Sec. 3.2. GNN relies on message passing as its core mechanism, enabling solutions to a wide range of graph applications (Wu et al., 2020). Therefore, we begin by breaking down the message passing into two *independent* operations: propagation (**P**) and transformation (**T**) (Zhang et al., 2022a). In short, the **P** step is a form of Laplacian smoothing that aggregates the information of neighbors, while the **T** step applies non-linear transformations to capture the data distribution of the training samples. On the one hand, in Sec. 4.2, theoretical analysis shows that the influence values

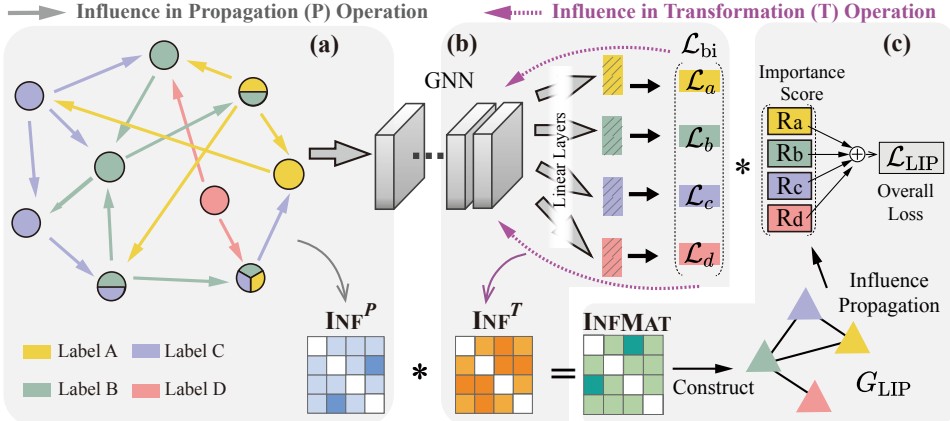

Figure 2: The overall framework of **LIP**: (a) quantify the influence correlations in **P** operation (Sec. 4.2) where the arrows with colors represents the mutual influence between labels; (b) quantify the influence correlations in **T** operation (Sec. 4.3) where $\mathcal{L}_{bi}$ separately calculates the loss of each label; (c) combine these two influence correlations to get the high-order label influence, and propagate through the constructed graph $G_{LIP}$ to calculate the important score $R$ for each label (Sec. 4.4).

in **P** operation are independent of the node embeddings, and therefore unrelated to the parameters of message passing. On the other hand, Sec. 4.3 analyzes the influence between labels in **T** operation which does not affect the topological structure between node sets with different labels, and is thus unrelated to the aggregation in message passing. Since the two operations are independent, the quantified influence can be further combined as the foundation of computing high-order influence correlations and help to improve MLNC task (Sec. 4.4).

The overall framework is shown in Fig. 2 with three main parts: (a) calculating the influence between labels in **P** operation, (b) calculating the influence between labels in **T** operation, and (c) propagating the influence on the label information propagation graph to form the final learning objective.

## 4.2  INFLUENCES IN PROPAGATION OPERATION

This *propagation operation* involves aggregating information from their contextual nodes, which may belong to different labels. Consequently, the intertwined distribution of nodes from various labels facilitates their interactions. During propagation, the interaction we seek to analyze is between two label sets $Y_a$ and $Y_b$ ($1 \leq a, b \leq k$), which is shown in Fig. 2(a). Specifically, we decompose the computation of influence correlations among labels during the P step into two parts: the magnitude of influence between all pairs of nodes ("Influence between node pairs") and the positive or negative direction of influence between node pairs with different labels ("Influence between label sets").

**Influence between node pairs.** We start the analysis by investigating the influence between node pairs $v_i$ and $v_j$. Inspired by previous work (Xu et al., 2018a; Wang & Leskovec, 2020; Zhang et al., 2021a), the influence from $v_i$ to $v_j$ can be quantified by measuring how alterations in the input feature $\mathbf{x}_i$ of $v_i$ affect the node embedding $\mathbf{z}_j$ of $v_j$ after $l$ iterations of message passing. In particular, for $v_i$ and $v_j$, supposing the message passing of the shared backbone $\Phi_\theta$ as $\mathbf{Z}^{(k)} = \hat{\mathbf{A}}^k \mathbf{Z}^{(0)} \theta$ where $\mathbf{Z}^{(0)} = \mathbf{X}$ and $\hat{\mathbf{A}} = \tilde{\mathbf{D}}^{-1/2} \tilde{\mathbf{A}} \tilde{\mathbf{D}}^{-1/2}$ is the symmetrically normalized adjacency matrix with self-loops, the influence of $v_i$ on the final embedding of $v_j$, *i.e.*, $I(i, j)$, can be defined as

$$I(i, j) = \frac{\partial \mathbf{z}_j^{(k)}}{\partial \mathbf{z}_i^{(0)}}. \tag{3}$$

Although the calculation of $I(i, j)$ initially involves node features, inspired by (Xu et al., 2018a), we prove that the magnitude of $I(i, j)$ is actually independent of the features themselves and is instead proportional in expectation to a random walk distribution $\pi_{lim}^{(ji)}$, namely $I(i, j) \propto \pi_{lim}^{(ji)}$. To further quantify and compute this distribution $\pi_{lim}$, inspired by (Gasteiger et al., 2019), we find that we can solving the Personalized PageRank (PPR) $\pi_{ppr}$ instead. By iteratively computing the PPR, we can obtain the influence correlations between any two nodes. Thus, we derive the quantification of the influence between any pair of nodes $v_i, v_j$ during the P phase as:

$$\text{INF}^P(v_i, v_j) := I(i, j) \propto \pi_{\text{ppr}}^{(ji)} = \{\alpha(\mathbf{I}_n - (1 - \alpha)\hat{\mathbf{A}})^{-1}\mathbf{s}_{v_i}\}_{v_j}, \tag{4}$$

where $\mathbf{I}_n \in \mathbb{R}^{n \times n}$ is the identity matrix, $\alpha \in (0, 1]$ is the teleport (or restart) probability, $\mathbf{s}_{v_i}$ is a one-hot indicator vector, and $\{\cdot\}_{v_j}$ means the $v_j$-th element of $\{\cdot\}$. Detailed discussion is in App. A.

**Influence between label sets.** Having computed the influence correlations magnitude $\text{INF}^P(v_i, v_j)$ between any pair of nodes during the P phase, represented as an $n \times n$ node influence correlations matrix, the second part involves integrating the influence from all nodes in $Y_a$ (with label $y_a$) on all nodes in $Y_b$ (with label $y_b$) to get the label influence correlation between $y_a$ and $y_b$. This part yields a $k \times k$ label influence relationship matrix $\text{INF}^P(y_a, y_b)$. A straightforward way would be to directly sum up the node level influence. However, as shown in our analysis in Fig. 1, there are both positive and negative influences between labels.

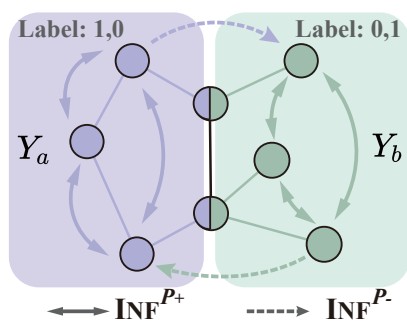

Figure 3: Illustration of the direction of positive and negative influence during P step. The nodes with both colors indicates the ones with both labels.

Therefore, we proceed to analyze the combinations of two node sets $Y_a$ and $Y_b$ with different labels to determine which node pairs exert positive influence and which exert negative influence. Specifically, if a node $v_i$ is labeled with $y_j$, it can be seen that this node is labeled as 1 in the binary classification of whether it has the label $y_j$; conversely, if it does not have the label $y_j$, it can be seen as being labeled as 0. Based on the Laplacian smoothing in the P operation (Zhang et al., 2022a), the information transmitted by source node $v_i$ labeled with 1 makes it easier for target nodes $v_j$ to be identified as labeled 1 in the same label space. Note that we break down the discussion of the influence direction between $Y_a$ and $Y_b$ by first analyzing the influence from $Y_a$ to $Y_b$. The reverse direction can be inferred by analogy. Thus, we first analyze the influence from $Y_a$ to $Y_b$, where the target is the set of all nodes with label $y_b$. From analysis in Fig. 3, we discuss the source from the following two aspects:

- In the label set $Y_a$, the nodes are labeled as $[1, 0]$ (possessing $y_a$), it therefore has negative influence on label set $Y_b$ ($[0, 1]$). However, the overlapping nodes are labeled as $[1, 1]$ where the positive and negative influences to target nodes are neutralized.

**Definition 3** (Source of Negative Influence $\text{INF}^{P-}$).

$$V_{neg} = Y_a \setminus (Y_a \cap Y_b) \rightarrow Y_b, \tag{5}$$

*where $\setminus$ means set difference and $\cap$ means intersection.*

- Intuitively, nodes with the label $y_b$, noted as $[*, 1]$, may contribute positive influence to all nodes in $Y_b$ (those with the label $[0, 1]$). However, nodes that simultaneously have both $y_a$ and $y_b$ labels (noted as $[1, 1]$) neutralize their positive and negative influences on $Y_b$. Therefore, the source of positive influence from $Y_a$ to $Y_b$ can be defined as follows:

**Definition 4** (Source of Positive Influence $\text{INF}^{P+}$).

$$V_{pos} = \overline{Y_a} \setminus \overline{Y_a \cup Y_b} = \overline{Y_a} \cap Y_b \rightarrow Y_b, \tag{6}$$

*where $\bar{\cdot}$ means complement and $\cup$ means union.*

Thus, by simply substituting $\text{INF}^{P+}$ as the index into Eq. 4 and normalizing by number of nodes, we can obtain the value of positive influence from $Y_a$ to $Y_b$; similarly, we can obtain the value of negative influence by substituting $\text{INF}^{P-}$. Adding the positive and negative influence together yields the complete influence in propagation operation $\text{INF}^P$ between any two labels.

## 4.3 Influences in Transformation Operation

In this section, we analyze interactions between label sets in transformation operation from message passing. The shared GNN backbone learns from the training data through model parameters in

this transformation operation. Moreover, the direction and magnitudes of parameter updates in the shared backbone $\Phi_\theta$ is determined by each label $y_i$, but different labels may require different directions and magnitudes of updates to reach their own optimal solutions. Thus, labels influence each other through parameter transformation during back propagation, which is through gradient descent. Next, we further quantify and analyze the gradient influence exerted by different labels.

Note that we use binary cross-entropy losses as $\mathcal{L}_{bi}$ for each label individually, which can be changed to other types of binary classification loss:

$$\mathcal{L}_{bi}(\mathbf{y}, \hat{\mathbf{y}}) = -\frac{1}{M} \sum_{i=1}^{M} \left[ \mathbf{y}_i \log(\hat{\mathbf{y}}_i) + (1 - \mathbf{y}_i) \log(1 - \hat{\mathbf{y}}_i) \right], \tag{7}$$

where $\mathbf{y} \in \{0, 1\}$ is a binary label indicating whether a node possess a certain label and $M$ is the number of the training nodes.

Therefore, the binary classification loss $\mathcal{L}_{bi}$ of each label exerts specific gradient $\nabla_{bi}$ to the shared component during the transformation operation:

$$\nabla_{bi} = \frac{\partial \mathcal{L}_{bi}}{\partial \Phi_\theta}. \tag{8}$$

These different gradients generated by the losses of different labels may either be mutually beneficial or mutually harmful. Therefore, during this transformation operation, any pair of labels will influence each other through their respective gradients as shown in Fig. 2(b).

Inspired by the definition of gradients conflicting (Yu et al., 2020), we propose that as the angle between the gradients of two losses increases, the positive influence between the labels becomes less significant. Conversely, the smaller the angle between the gradients generated by different labels, the smaller the negative influence they have on each other. Accordingly, here we give a formal definition on measuring the influence in transformation $\mathbf{T}$ step.

**Definition 5** ($\mathbf{T}$ step Influence $\text{INF}^T$). *Given two losses $L_a$ and $L_b$ ($1 \leq a, b \leq k$) from label $y_a$ and $y_b$ respectively, the influence in transformation operation is the accumulated differences between the gradients $\nabla_a$ and $\nabla_b$:*

$$\text{INF}^T(a, b) = \text{INF}^T(b, a) = -\sum_\theta \text{ANGLE}(\nabla_a(\theta), \nabla_b(\theta)), \tag{9}$$

*where $\nabla_a(\theta)$ is the gradient of binary loss $\mathcal{L}_a$ on shared backbone $\Phi_\theta$, and $\text{ANGLE}(\cdot)$ can measure the angle between directions of two gradients. In our implementation, we determine the angle between the two gradients by calculating the cosine similarity.*

### 4.4 INFLUENCE PROPAGATION ON LABEL GRAPH

**Label graph construction.** From the previous work (Zhang et al., 2022a; Wu et al., 2019; Zhang et al., 2022b) and the above analysis, it can be seen that the influence between labels during propagation and transformation operation is independent of each other. Therefore, the influences from these two parts can be combined through multiplication to obtain the final relationship matrix of label influence:

$$\text{INFMAT} = \text{INF}^P * \text{INF}^T, \quad \text{INFMAT} \in \mathbb{R}^{k \times k}, \tag{10}$$

where $*$ means element-wise multiply. Each value in INFMAT reflects the relationship between each pair of labels. However, we still need to analyze the higher-order correlations between labels.

Thus, we propose to build a *label graph* $G_{LIP}$. Note that INFMAT contains both positive and negative values. Our goal is to enhance the positive influence and suppress the negative influence between labels. Thus, we introduce a simple method yet can preserve higher-order influence to convert the INFMAT into label graph $G_{LIP}$:

**Definition 6.** *Given the label influence matrix $\text{INFMAT} \in \mathbb{R}^{k \times k}$, we define label graph as*

$$G_{LIP} = (\mathbf{V}_y, \mathbf{A}_{LIP}), \quad \text{with } \mathbf{A}_{LIP} = \text{SOFTMAX}_{row}(\text{INFMAT}), \tag{11}$$

*where $\text{SOFTMAX}_{row}(\cdot)$ represents calculating softmax along the rows which means that we tend to treat the negative influence as weak positive influence in the higher-order view. $\mathbf{V}_y$ denotes the label nodes (triangle nodes in Fig. 2(c)) which represent a certain label type. $\mathbf{A}_{LIP}$ is the weighted adjacent matrix denoting the overall higher-order correlations between all the labels.*

**Overall loss function.** Intuitively, if a label is detrimental to many other labels, we suppressed it by reducing its coefficient of the loss corresponding to this label to lessen its negative impact. Conversely, if a label is beneficial to many other labels, we encourage it by increasing the coefficient of the loss to enhance its positive impact. Thus, we propose to quantify the importance coefficient (score) based on the label graph $G_{LIP}$. Since the higher-order influence propagation take place on the graph $G_{LIP}$, we turn to the computation of PageRank (Page et al., 1999) for $G_{LIP}$.

Formally, we calculate the importance score $R \in [0,1]^k$ of each label node as

$$R = (\mathbf{I}_h - \beta \mathbf{A}_{LIP})^{-1} \left( \frac{1-\beta}{h} \right) \mathbf{1}, \tag{12}$$

where $\mathbf{I}_h$ is the identity matrix, $\beta \in (0,1)$ is the teleport (or restart) probability here, $\mathbf{1}$ is an $h$-dimensional vector in which all elements are equal to 1.

Incorporating this importance coefficient $R$, we present the overall loss function as

$$\mathcal{L}_{\mathbf{LIP}} = \sum_{j=0}^{k} R_j \mathcal{L}_{\text{bi}}(\hat{\mathbf{y}}_j, \mathbf{y}_j), \tag{13}$$

where $\mathcal{L}_{\text{bi}}$ is from Eq. 7 which can be changed to other imbalanced classification loss (Chen et al., 2024b; Zhuo et al., 2024).

## 5 EXPERIMENTS

In this section, we evaluate **LIP** and aim to answer the following research questions:

- **RQ1.** How does **LIP** perform on MLNC task with different settings and label ratios?
- **RQ2.** Can **LIP** serve as a plug-and-play booster to be equipped with any backbone?
- **RQ3.** How reliable are the label influence relationships we proposed to capture?

### 5.1 EXPERIMENTAL SETTINGS

To comprehensively validate our framework, we conduct experiments on 2 classical MLNC datasets (DBLP (Akujuobi et al., 2019), BlogCat (Shi et al., 2020a)), 1 large scale OGB dataset (Ogbn-proteins (Hu et al., 2020), OGB-p in short), and 3 new biological datasets (PCG, HumLoc, Euk-Loc (Zhao et al., 2023)) from different domains. The statistics of datasets are in App. D. We use three types of baselines for comparison. First, we combine unsupervised embedding and multi-label classifier. We adopt the frequently used unsupervised graph embedding methods Node2Vec (Grover & Leskovec, 2016), the classic multi-label classifiers binary relevance (BR) (Tsoumakas et al., 2010) and classifier chain (CC) (Read et al., 2009). Second, we combine semi-supervised backbone with SOTA multi-label methods. We adopt the same backbone for baselines and our own for fairness. We choose GCN (Kipf & Welling, 2017) as main backbone, ML-KNN (Zhang & Zhou, 2007) and latest work PLAIN (Wang et al., 2023) as multi-label methods. The third group is backbone-relevant methods which specifically designed for MLNC on graphs. We choose MLGW (Akujuobi et al., 2019), ML-GCN (Gao et al., 2019), LARN (Xiao et al., 2022), LANC (Zhou et al., 2021) and the latest work VariMul (Song et al., 2021). Furthermore, "backbone+Auto" means the coefficients of losses of different labels are learned automatically. More details are in App. E.1.

### 5.2 COMPARISON WITH BASELINES

To answer **RQ1**, we evaluate **LIP** against various kinds of baselines on classic and latest benchmark datasets, ranging from multi-label techniques used on any data form to methods tailored for MLNC on graphs. To fully evaluate the effectiveness against baselines, we adopt 2 split settings: node split and label split. Refer to App. E.1 to see the details and differences between them. We also evaluate under different training ratio (App. E.2).

From Tab. 1, we can draw several conclusions. First and foremost, **LIP** is the most effective one most of the time, which verifies the effectiveness of our approach. Our method outperforms other baselines by 3.06% on AUC and 2.54% on AUC on average. Note that our methods can boost the performance based on the backbones. Second, we find that while other methods including VariMul,

Table 1: Performance (mean ± std. deviation) under node split at 6:2:2 in terms of Macro AUC, AP.

| Metrics | DBLP Macro AUC | AP | BlogCat Macro AUC | AP | OGB-p Macro AUC | AP |
|---|---|---|---|---|---|---|
| Node2vec+BR | 71.22 ± 0.31 | 57.41 ± 1.29 | 53.76 ± 1.08 | 6.56 ± 0.75 | 50.22 ± 3.01 | 1.92 ± 0.73 |
| Node2vec+CC | 72.57 ± 0.24 | 57.13 ± 1.88 | 57.97 ± 1.31 | 8.93 ± 0.82 | 51.39 ± 2.14 | 2.07 ± 0.31 |
| GCN+ML-KNN | 90.11 ± 1.02 | 80.01 ± 0.77 | 60.18 ± 2.33 | 9.16 ± 0.60 | 55.07 ± 2.73 | 2.02 ± 0.33 |
| GCN+PLAIN | 80.55 ± 1.23 | 73.44 ± 1.12 | 63.95 ± 1.85 | 10.21 ± 0.56 | 66.29 ± 3.45 | 3.26 ± 0.67 |
| MLGW | 73.32 ± 1.44 | 56.03 ± 0.47 | 60.02 ± 2.19 | 9.81 ± 0.46 | 50.70 ± 2.54 | 1.78 ± 0.73 |
| ML-GCN | 72.66 ± 2.73 | 56.71 ± 2.59 | 60.97 ± 2.21 | 9.68 ± 0.39 | 60.11 ± 2.65 | 2.19 ± 0.54 |
| LARN | 74.29 ± 2.53 | 58.11 ± 1.25 | 63.18 ± 1.84 | 9.77 ± 0.63 | 68.18 ± 1.33 | 3.08 ± 0.16 |
| LANC | 91.68 ± 0.42 | 83.50 ± 0.83 | 67.94 ± 3.30 | 10.35 ± 0.82 | 68.75 ± 0.51 | 4.11 ± 0.23 |
| VariMul | 92.14 ± 1.23 | 85.30 ± 0.92 | 68.71 ± 2.97 | 13.74 ± 0.83 | 70.73 ± 1.67 | 2.31 ± 0.44 |
| GCN+Auto | 92.13 ± 1.57 | 85.48 ± 1.32 | 66.05 ± 1.25 | 12.53 ± 0.51 | 71.39 ± 1.34 | 2.11 ± 0.22 |
| **GCN+LIP** | **94.38 ± 1.51** | **87.45 ± 1.28** | **70.21 ± 2.02** | 12.33 ± 0.91 | **74.82 ± 0.34** | **5.72 ± 0.28** |

| Metrics | PCG Macro AUC | AP | HumLoc Macro AUC | AP | EukLoc Macro AUC | AP |
|---|---|---|---|---|---|---|
| Node2vec+BR | 52.66 ± 1.73 | 15.13 ± 0.25 | 54.17 ± 1.58 | 10.67 ± 0.42 | 51.29 ± 1.60 | 6.00 ± 0.72 |
| Node2vec+CC | 52.92 ± 0.62 | 14.54 ± 0.51 | 54.81 ± 1.29 | 11.53 ± 0.36 | 52.18 ± 1.16 | 6.55 ± 0.59 |
| GCN+ML-KNN | 56.11 ± 0.50 | 18.60 ± 0.82 | 57.99 ± 0.35 | 11.76 ± 0.77 | 63.95 ± 1.15 | 9.34 ± 0.77 |
| GCN+PLAIN | 59.75 ± 1.70 | 20.16 ± 1.00 | 60.15 ± 1.28 | 14.71 ± 1.51 | 68.02 ± 1.10 | 11.72 ± 0.69 |
| MLGW | 55.86 ± 3.92 | 15.59 ± 1.33 | 56.92 ± 1.03 | 10.54 ± 0.43 | 51.77 ± 1.41 | 6.02 ± 0.55 |
| ML-GCN | 57.24 ± 2.44 | 19.45 ± 1.13 | 60.79 ± 1.38 | 14.55 ± 0.46 | 54.17 ± 1.88 | 6.92 ± 0.49 |
| LARN | 57.79 ± 0.61 | 19.09 ± 0.38 | 61.48 ± 1.22 | 17.89 ± 0.62 | 57.79 ± 1.28 | 7.41 ± 0.80 |
| LANC | 56.58 ± 0.63 | 19.51 ± 0.93 | 59.63 ± 1.21 | 13.26 ± 0.98 | 51.25 ± 0.63 | 6.11 ± 0.89 |
| VariMul | 62.77 ± 0.34 | 21.93 ± 0.52 | 67.42 ± 2.44 | 23.34 ± 0.88 | 71.44 ± 1.66 | 13.74 ± 0.76 |
| GCN+Auto | 58.53 ± 0.47 | 19.11 ± 0.83 | 66.07 ± 1.17 | 21.58 ± 1.09 | 70.25 ± 1.25 | 13.05 ± 0.46 |
| **GCN+LIP** | **65.73 ± 0.52** | **22.97 ± 1.69** | **73.22 ± 1.76** | **25.18 ± 1.16** | **72.92 ± 1.82** | **15.26 ± 0.56** |

LARN, and PLAIN achieve good results, methods tailored for MLNC on graph data always outperform those designed for all data form. The reason is that integrating multi-label processing into the encoding of graph structure and node features makes the node embeddings more informative, while methods that separate node encoding from multi-label correlation mining struggle to achieve optimal results. Third, the methods using unsupervised node embedding perform even worse since there is no label information when representing the nodes. Although methods like "backbone+Auto", which automatically learn loss coefficients, appear more flexible, setting these coefficients solely based on the loss is like searching for a needle in a haystack, making it difficult to amplify the loss of beneficial labels during the learning process. On one hand, our method better explores the relevant impact of the nodes contained by labels from the perspectives of both message passing and gradient update; on the other hand, during the training process, our method is also constantly related to the current state of the model. This enables our approach to dynamically mine the relationships of nodes with different labels in the process of forward and backward information transmission, thereby making the model more effective.

## 5.3 COMPARISON WITH DIFFERENT BACKBONES

One of the advantages of **LIP** is the ability of being plugged into any node embedding methods. Thus, to answer **RQ2**, we propose to change the backbone to others with different inductive bias. Here we compare our method with the "Com", which directly sums the multi-label loss to train the model. We also change backbones on more datasets in App. E.2. Since some of the datasets are heterophily graphs (see Tab. 4), we use backbones designed for homophily and heterophily graphs.

**Results.** We change backbones on BlogCat dataset in Fig. 4(a). We can see that **LIP** constantly achieves large improvement when applied to any backbone, demonstrating the effectiveness of our approach. Although BlogCat is a heterophily graph, H2GCN does not perform well comparing to others. Similar results can be found in (Zhao et al., 2023). However, H2GCN perform best on Abnormal as a heterophily graph in App. E.2. We believe there are two reasons for this. First, BlogCat has no node features, so we used Node2Vec as feature for all methods, making the input of backbone actually a homogeneous graph from a feature perspective. However, H2GCN's strategy of aggregating more neighbor features makes it harder for nodes to learn the labels. Second, the heterophily here is calculated from multi-label, but H2GCN is designed for single-label, so it may

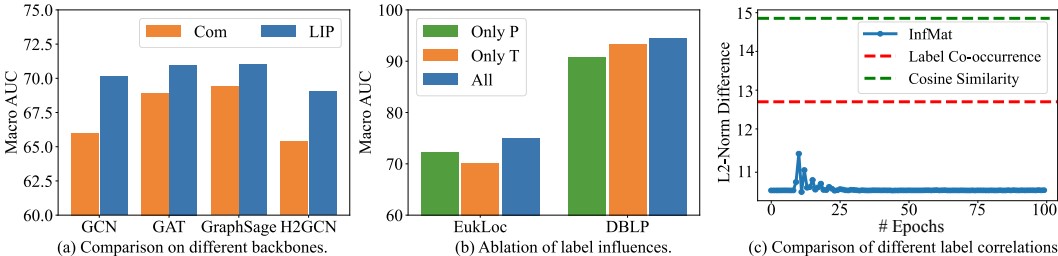

Figure 4: Model Analysis showing the effectiveness of **LIP**.

not perform better. An interesting observation is that different backbone models exhibit varying degrees of sensitivity to our method. For instance, GCN show greater improvements before and after the application of our method compared to GAT. We think that this may be due to our method's relation to gradient calculation, with different models having varying sensitivities to labels.

### 5.4 DISCUSSION ON LABEL CORRELATIONS

To answer the **RQ3**, we evaluate the label influence correlation mined by **LIP**. First, we decompose the proposed two parts of influence correlations to evaluate effectiveness of each correlation separately. Then, we compare our method with other commonly used approaches for explicitly computing correlations to see which more accurately reflects the influence correlations between labels.

**Ablation study on correlations.** The "Only P", "Only T" and "All" means using $\text{INF}^P$, $\text{INF}^T$ and INFMAT respectively, where GCN is the backbone. As shown in Fig. 4(b), the performance using the whole INFMAT is the best, which prove the utility of the combination of the influence during both forward and backward propagation. Moreover, we find that $\text{INF}^P$ perform better than $\text{INF}^T$ on EukLoc while the other way around on DBLP. We believe the reason is that EukLoc is more heterophilic, making it more dependent on global graph structure. Our $\text{INF}^P$ focuses on uncovering influence during the propagation process, which emphasizes the structural aspects more during training compared to $\text{INF}^T$. More ablation study on normalization is in App. E.2.

**Compare with other label correlations.** Here, we can verify if our captured label correlation align with expectations. Specifically, we aim to capture the pairwise label correlations similar as shown in Fig. 1, which contain positive and negative correlations. Note that we cannot use this performance differences as the label correlations in the model, as retraining the performance to calculate them are not possible. Therefore, we compare which method can obtain the relationship matrix that most closely matches the influence relationships inferred from the results. Here we collect all methods used for explicitly calculating multi-label correlations: co-occurrence from (Wang et al., 2023) and cosine similarity of label embeddings from (Xu et al., 2018b). We calculate the *l2*-norm difference between the three methods we compare and the influence correlations as epochs increase (with GCN backbone on DBLP). As shown in Fig. 4(c), although INFMAT changes over epochs, the difference between INFMAT and ground truth correlation is the closest. Moreover, it can be observed that the label co-occurrence measurement is closer compared to the cosine similarity of labels. Additionally, for a significant portion of epochs, the distance relative to ground truth does not change much, not because the influence we measure is static, but because the influence between different pairs of labels waxes and wanes, yet always remains close to the ground truth. Due to space limit, we place the results of other experiments in the App. E.

### 6 CONCLUSIONS

In this paper, we address the challenges of the valuable but overlooked MLNC task by introducing a novel approach to understanding and leveraging label influence correlations on graphs. By decomposing the influences between labels in message passing mechanism, we provide insights on analyzing and quantifying label influences during the learning process. Our proposed method, **LIP**, boosts the MLNC by ensuring that positive label influences are amplified while mitigating the negative ones. Empirical results across comprehensive MLNC datasets demonstrate the effectiveness of **LIP**, achieving consistent improvements in various metrics.

ACKNOWLEDGMENTS

This research is supported by the National Research Foundation, Singapore and Infocomm Media Development Authority under its Trust Tech Funding Initiative, and the National Research Foundation, Singapore under its AI Singapore Programme (AISG Award No: AISG2-TC-2021-002). Any opinions, findings and conclusions or recommendations expressed in this material are those of the author(s) and do not reflect the views of National Research Foundation, Singapore and Infocomm Media Development Authority. This work is also supported by NSFC (No. 62322606, No. 62441605), Zhejiang NSF (LR22F020005), and CCF-Zhipu Large Model Fund.

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

APPENDIX

# A   DISCUSSIONS ON INFLUENCE FROM PROPAGATION

This section aims to supplement Sec 4.2 regarding the pair wise influence value from propagation (**P** operation). Specifically, inspired by previous work (Xu et al., 2018a; Wang & Leskovec, 2020), we prove the correctness of the calculation method on influence value between nodes with different labels in propagation operation.

**Theorem 1** (Influence Value in Propagation). *During the propagation phase in message passing, the influence value between two nodes with different labels is equivalent, in expectation, to the transition probability between them as calculated by Personalized PageRank (PPR) (Gasteiger et al., 2019; Chen et al., 2021). Namely, the influence value between $v_i, v_j$ is*

$$\text{INF}^P(v_i, v_j) = \{\alpha(I_n - (1-\alpha)\hat{\mathbf{A}})^{-1}s_{v_i}\}_{v_j} \propto I(i, j), \tag{14}$$

*where $\mathbf{I}_n \in \mathbb{R}^{n \times n}$ is the identity matrix, $\alpha \in (0, 1]$ is the teleport (or restart) probability, $\mathbf{s}_{v_i}$ is a one-hot indicator vector and $\{\cdot\}_{v_j}$ means the $v_j$-th element of $\{\cdot\}$.*

*Proof.* We use $k$ layers of GCN Kipf & Welling (2017) as backbone $\Phi_\theta$ for analysis. We first prove the the influence value between $v_i, v_j$ with different labels is equivalent, in expectation, to the $k$-step random walk distribution on graph $G$ starting at node $v_i$.

From the definition of influence from $v_i$ to $v_j$ in Eq. 3, we proceed with the detailed calculations:

$$I(v_i, v_j) = \frac{\partial z_j^{(k)}}{\partial z_i^{(0)}} = \frac{1}{\widetilde{\deg(v_i)}} \cdot \text{diag}\left(1_{f_{v_i}^{(k)} > 0}\right) \cdot W_k \cdot \sum_{z \in \widetilde{N}(v_i)} \frac{\partial z_j^{(k-1)}}{\partial z_i^{(0)}}, \tag{15}$$

where $f_{v_i}^{(k)}$ is the pre-activated embedding of $z_j^{(k)}$, $W_k$ stands for the last layer weight matrix of GCN. Furthermore, we use chain rule to calculate any rank $l$ of the derivative as

$$\frac{\partial z_j^{(l)}}{\partial z_i^{(0)}} = \sum_{p=1}^{\lambda} \left[\frac{\partial z_j^{(l)}}{\partial z_i^{(0)}}\right]_p = \sum_{p=1}^{\lambda} \prod_{m=l}^{1} \frac{1}{\widetilde{\deg\left(v_p^m\right)}} \cdot \text{diag}\left(1_{f_{v_p^m}^{(m)} > 0}\right) \cdot W_m, \tag{16}$$

where $\lambda$ is the total number of paths of length $l+1$ from node $v_j$ to $v_i$. For certain path $p$, $v_p^m$ is node $v_i$, $v_p^0$ is node $v_j$ and for $m = 1...(l-1)$, $v_p^(k-1) \in \widetilde{N}(v_p^m)$. More specifically, we can calculate an entry of the derivative as

$$\left[\frac{\partial z_j^{(l)}}{\partial z_i^{(0)}}\right]_p^{(a,b)} = \prod_{l=k}^{1} \frac{1}{\widetilde{\deg\left(v_p^m\right)}} \sum_{q=1}^{\gamma} R_q \prod_{m=l}^{1} w_q^{(m)}, \tag{17}$$

where $\gamma$ is the number of paths $q$ from the $a$ to $b$ in the computation directed acyclic graph (DAG) of $\left[\frac{\partial z_j^{(l)}}{\partial z_i^{(0)}}\right]_p$. For each layer $m$, $w_q^{(m)}$ is the entry of $W_m$ which is used in the $q$-th path. $R_q \in {0,1}$ denote whether the $q$-th path is active or not as the result of ReLU activation of entries of $f_{v_p^m}^{(m)}$'s on the $q$-th path. Since the variable $R$ follows a Bernoulli distribution, for all $q$, $Pr(R_q = 1) = \rho$, we have

$$\mathbb{E}\left[\left[\frac{\partial z_j^{(l)}}{\partial z_i^{(0)}}\right]_p\right] = \rho \cdot \prod_{m=l}^{1} \frac{1}{\widetilde{\deg\left(v_p^m\right)}} \cdot w_q^{(m)}. \tag{18}$$

Thus, we know that

$$\mathbb{E}\left[\frac{\partial z_j^{(l)}}{\partial z_i^{(0)}}\right] = \rho \cdot \prod_{m=l}^{1} W_m \cdot \left(\sum_{p=1}^{\lambda} \prod_{m=l}^{1} \frac{1}{\widetilde{\deg\left(v_p^m\right)}}\right). \tag{19}$$

On the other hand, $k$-step random walk probability at $v_i$ can be calculated by summing up the probability of all paths of length $k$ from $v_i$ to $v_j$, which is exactly $\sum_{p=1}^{\lambda} \prod_{m=l}^{1} \frac{1}{\widetilde{\deg(v_p^m)}}$. Moreover,

the random walk probability starting at $v_i$ to the other nodes sum up to 1. By multiplying and normalizing, the conclusion can be immediately derived.

Moreover, the distribution $\pi$ of random walk with restart is $\pi^{(t+1)} = \alpha v + (1-\alpha)A\pi^{(t)}$ in iteration. We can derive the solution (Gasteiger et al., 2019) as $\pi = \alpha \left(I - (1-\alpha)A\right)^{-1} v$, which is the same as the calculation equation in Eq. 4. $\qquad\square$

## B  TIME COMPLEXITY COMPARISON

### B.1  THEORETICAL COMPARISON

We now analyze the time complexity of **LIP** based on vanilla MLNC (Sec. 3.2). The calculation of influence in Sec. 4.2 (**P** step) is part of data pre-processing where we can use fast personalized PageRank algorithms designed for large graphs. Thus, the additional time includes two parts: (i) the influence calculation in transformation operation which is the same as the forward calculation of GNN, namely $O(nfd + ed)$, where $f$ is feature dimension and $d$ is hidden dimension of GNNs, $n$ and $e$ are the number of nodes and edges in original graph; (ii) the important score calculation from $G_{LIP}$ costs $O(t \cdot e_{LIP})$, where $t$ is the number of iterations and $e_{LIP}$ is the number of influence edges in $\mathbf{A}_{LIP}$. Thus, the overall time complexity is $O(n + e + t \cdot e_{LIP})$ since $f, d \ll n, e$.

We select several representative baselines and analyzed their time complexities, as shown in the Tab. 2 below.

Table 2: The comparison of time complexity between **LIP** and other baselines.

| Method | GCN+ML-KNN | MLGW | LANC |
|---|---|---|---|
| Complexity | $O(n \cdot \log n + k \cdot l + e)$ | $O(n^3 + e \cdot f^2)$ | $O(n + e + n \cdot b^2 \cdot f)$ |
| Method | VariMul | GCN+Auto | **LIP** |
| Complexity | $O(n + e + l \cdot \log l)$ | $O(n + e)$ | $O(n + e + t \cdot e_{LIP})$ |

In the table above, $k$ represents the hyperparameter for the k-nearest neighbors algorithm, $l$ stands for the number of labels, $b$ denotes the number of sample neighbors set in LANC, and $f$ refers to the size of its hidden layer. The table shows that our method has almost the lowest time complexity, and experiments confirm that it can scale to large graph datasets effectively.

### B.2  EMPIRICAL COMPUTATIONAL COST

We add a table below showing the average per-epoch training time and GPU memory usage of various methods on graph data from two different domains.

Table 3: Computational cost of each methods, including time and space during training.

| | Cost | ML-KNN | MLGW | LANC | VariMul | GCN+Auto | GCN+LIP |
|---|---|---|---|---|---|---|---|
| DBLP | Time (s/epoch) | 0.012 | 5.710 | 2.350 | 1.015 | 0.003 | 0.008 |
| BlogCat | Time (s/epoch) | 0.100 | 9.710 | 3.837 | 2.082 | 0.051 | 0.311 |
| DBLP | GPU mem (MB) | 2200 | 3530 | 2204 | 2980 | 2271 | 1595 |
| BlogCat | GPU mem (MB) | 3010 | 5036 | 2800 | 3572 | 3073 | 2369 |

As observed, our method aligns with the conclusions analyzed in Appendix B, demonstrating relatively shorter training times and especially lower GPU memory usage. This indicates that our method has better scalability. Specifically, our training time is roughly on the same order of magnitude as a standard GCN but takes a few times bigger. However, compared to other baselines, our method typically requires one order of magnitude less time. On the other hand, since the quantification of influence correlations during the T step involves gradient calculations, our method essentially splits the gradient calculation into k smaller steps. This trades off some computation time for reduced memory usage, allowing our method to achieve even lower GPU memory consumption than a standard GCN.

## C  EXTENDED RELATED WORK

**Multi-label correlation mining.**  Exploring multi-label correlation in other data forms, e.g. computer vision (CV), is also a key challenge. These methods can be divided into three categories: first order, second order and high-order (Zhang & Zhang, 2010). First-order methods disregard label correlations, assuming all labels are independent. Binary Relevance (BR) (Tsoumakas et al., 2010), transforms the problem into separate binary classification tasks. Similarly, ML-KNN (Zhang & Zhou, 2007) is based on k-nearest-neighbor classification. While these methods are highly efficient, their neglect of label correlations leads to reduced performance. Second-order methods emphasize pairwise label correlations. Approaches like calibrated label ranking (CLR) (Fürnkranz et al., 2008) transform into pairwise ranking problems. While they are generally more effective than first-order approaches at exploiting label correlations, label relationships can be highly complex in the real-world applications. High-order methods aim to explore higher-order label correlations. Classifier chains (CC) (Read et al., 2009) trains a chain of binary classifiers, each predicting the current label based on the features and previously predicted labels. LACO (Zhang et al., 2021b) introduces two additional losses for label correlation prediction. HOMI (Si et al., 2023) argue that the label matrix is approximately full-rank and use the label correlation to regularize the prediction which is similar to PLAIN (Wang et al., 2023). However, these methods do not analyze the complex non-Euclidean nature of graphs. For image and text data, where data points are independent, label correlations exist within label semantics. For graph data, the topological structure between node sets with different labels makes the analysis more challenging. Moreover, as shown in Fig. 1, we find that MLNC on graphs requires uncovering the *influence correlations* rather than the proximity correlations explored in other works. In other words, there can be both positive and negative influences between labels on graphs, while previous studies could only capture positive label proximity.

**Label Propagation Methods.**  The Label Propagation Algorithm (LPA) is a classic method whose application extends even beyond graph-based tasks. The fundamental assumption of LPA is that labels vary smoothly over the edges of a graph. As a result, given a graph and the labels of some of its nodes, LPA can infer the labels of the remaining nodes. However, traditional LPA does not take advantage of node feature information. This limitation has inspired many efforts to incorporate its principles or combine its ideas with Graph Neural Networks (GNNs) to construct powerful models. Some work leverage the simplicity and effectiveness of the LPA to assist graph models for node classification. GMNN Qu et al. (2019) models the joint distribution of labels with node features using Conditional Random Fields (CRF) Lafferty et al. (2001) and employs a variational EM framework Neal & Hinton (1998) for efficient training. In the E-step, a GNN learns representation to approximate the posterior distribution of labels. In the M-step, another GNN models the dependencies between labels. The M-step is quite similar to LPA, except that it is learnable and nonlinear. C&S Huang et al. (2020) proposes combining shallow models that ignore graph structure with two post-processing methods including error correction (correct) and label propagation (smooth), resulting in significant performance improvements with a simple model. GCN-LPA Wang & Leskovec (2021) theoretically analyzes the relationship between GCN and LPA, proving that edge weights that enhance LPA can also improve GCN. Based on this insight, they proposed the GCN-LPA model, which incorporates LPA as a regularization term for GCN. Another line of work uses LPA to correct noisy labels. LPM Xia et al. (2021) utilize the intrinsic graph structure to propagate labels and combines meta-learning to aggregate labels. Furthermore, R2LP Cheng et al. (2024) extends the use of LPA to arbitrary heterophily levels, simultaneously propagating labels to unlabeled nodes and correcting noisy labels. R2LP not only generalizes LPA to more realistic scenarios involving heterophily graphs and varying noise levels but also provides a theoretical analysis of its effectiveness on label denoising.

The differences between our Label Influence Propagation (LIP) and the LPA-related works are threefold: Propagation target: We do not propagate labels; instead, we propagate the influence between labels. Propagation medium: In LPA, the propagation medium is typically the graph structure that connects nodes. In contrast, the propagation medium in LIP is the label propagation graph, constructed by quantified pairwise label influences. In this graph, nodes are labels, and edges are the influence correlations between labels. Purpose of propagation: The purpose of propagating label influence is not to infer unknown labels or denoise existing labels, but rather to extend the computed pairwise label influences to high-order label influences. This ultimately encourages labels with positive influence and suppresses those with negative influence.

## D  DATASETS

To validate the proposed framework, we conduct experiments on 8 datasets, including 3 classical MLNC datasets (Delve-M (Akujuobi et al., 2019), DBLP (Akujuobi et al., 2019), BlogCat (Shi et al., 2020a)), 1 private dataset (Abnormal), 1 large scale OGB dataset (Ogbn-proteins (Hu et al., 2020), OGB-p in short), 3 new biological datasets (PCG, HumLoc, EukLoc (Zhao et al., 2023)) from different domains. The private one is a real-world anonymous graph multi-label anomaly detection dataset under the premise of ensuring data privacy. At present, the data Abnormal is not yet publicly available, but it may be released in the form of benchmarks in the future. The statistics of these datasets are in following Tab. 4.

**BlogCat.**  The data lacks inherent node features, so feature initialization is required. Although the paper (Zhao et al., 2023) uses an identity matrix to initialize node features on BlogCat, their experimental results show that DeepWalk (Perozzi et al., 2014) significantly outperforms other methods using identity matrix initialization. Thus, we use DeepWalk embedding as node feature of BlogCat in Tab. 1 and Fig. 4. Moreover, we find that another frequently used feature initialization method on BlogCat is to use structure-base embedding (Qiu et al., 2020). Hence we use their structure-based initialization method in Tab. 6 and Tab. 7.

Table 4: The statistics of all the MLNC grap datasets.

|  | DBLP | BlogCat | OGB-p | PCG |
|---|---|---|---|---|
| # nodes | 28,702 | 10,312 | 132,534 | 3,233 |
| # edges | 68,335 | 667,966 | 39,561,252 | 37,352 |
| # labels | 4 | 39 | 112 | 15 |
| $r_{homo}$ | 0.76 | 0.10 | 0.15 | 0.17 |
| Domain | Citation Network | Social Network | PPI | PPI |
| Node | Author | Blogger | Protein | Protein |
| Edge | Co-authorship | Friendship | Biological associations | Biological associations |
| Label | Research areas | Interests | Functions | Phenotypes |
|  | HumLoc | EukLoc | Delve-M | Abnomal |
| # nodes | 3,106 | 7,766 | 1,229,280 | $0.1 \sim 10M$ |
| # edges | 18,496 | 13,818 | 4,322,275 | $10 \sim 100M$ |
| # labels | 14 | 22 | 20 | 11 |
| $r_{homo}$ | 0.42 | 0.46 | 0.65 | $0.05 \sim 0.20$ |
| Domain | PPI | PPI | Wikipidia Network | Anomaly detection |
| Node | Protein | Protein | Paper | User |
| Edge | Biological associations | Biological associations | Citation | – |
| Label | Subcellular locations | Subcellular locations | Topics | Anomaly |

## E  MORE EXPERIMENTS

### E.1  SETTINGS AND DETAILS

**Reproducibility**  Experiments are conducted using 2 NVIDIA 3090 GPUs. Each experiment is replicated five times, utilizing different seeds for each run to ensure robustness and reproducibility. We reproduce the baselines by the released code or corresponding description in their paper. And we follow the hyper-parameter setting suggested in their papers to ensure the fairness of comparison. The implementation settings and details are uploaded in the anonymous repository: `https://github.com/Xtra-Computing/LIP_MLNC`.

**Evaluation metrics.**  Macro AUC and AP are commonly used non-parametric metrics in prior works. Macro AUC is the area under the ROC curve, and it provides an aggregate measure of performance across all possible classification thresholds, reflecting the model's ability to distinguish between positive and negative classes. AUC is the official metric provided for comparison on the OGB leaderboard (Hu et al., 2020), and it is also used in benchmarks (Tang et al., 2023) for highly imbalanced scenarios. Moreover, as noted in (Zhao et al., 2023; Yang et al., 2015; Sun et al., 2025; 2024), AUC can be misleading for highly imbalanced datasets. (Zhao et al., 2023) also reveals that

OGB-p has a nearly 90% of unlabeled nodes, indicating extreme imbalance for most labels. Thus, we also report Macro F1 and AP. Macro F1 computes the F1 score for each class independently and then takes the average. AP (Average Precision) is a performance metric that summarizes the precision-recall curve by computing the weighted mean of precisions at different thresholds. It provides an overall measure of a model's ability to balance precision and recall, particularly useful in imbalanced datasets. Higher values of these metrics indicates superior performance of **LIP**.

Note that AP, F1 and AUC can show significant differences due to their focus on different aspects of classification performance. F1 is sensitive to the balance between precision and recall, making it crucial for imbalanced datasets. AUC evaluates the overall ability to discriminate between classes across all thresholds. Therefore, a model might have a high AUC but a lower F1 if it struggles with precision or recall for the positive class. Moreover, AUPR (Area Under the Precision-Recall Curve) is also a well-known classification metric. However, AUPR may overestimate model performance when the number of thresholds or unique prediction values is limited. Hence we omit this metric.

**Hyper-parameter settings.** For training process, we use Adam optimizer with early stopping at 100 epochs to train **LIP**. Moreover, other hyper-parameters are decided using random search strategy and the range of hyper-parameters are listed in Tab. 5. When comparing with other baselines, we set the same number of layers for the backbone if the same backbone is used.

**Backbones.** As shown in the Tab. 4, our datasets include both homophily and heterophily datasets, so we used four different backbones to validate the effectiveness of our method. Among them, GCN (Kipf & Welling, 2017), GAT (Velicković et al., 2017), and GraphSage (Hamilton et al., 2017) are commonly used backbones for homophily graphs, while H2GCN (Zhu et al., 2020) is a popular model for heterophily graphs.

Table 5: The hyperparameter setting in this paper for all datasets

| Hyper-parameter | RangeValue |
|---|---|
| Hidden size | {32,64,128,256,512} |
| Learning Rate | {1e-3 → 5e-1} |
| Weight decay | {1e-2,5e-3,1e-4,5e-4,1e-5,5e-6,1e-7,0} |
| Dropout rate | {0 → 0.8} |
| Optimizer | Adam |
| Epoch | 1000 |
| Early stopping patience | 100 |

### E.2 EVALUATION UNDER LABEL SPLIT

**Two split settings.** We employ two types of split settings to show the robustness of our method on MLNC task. The first setting is node split, where the dataset is divided into training, validation, and test sets based on nodes, meaning that every node in the training set has full label information. This corresponds to the common single-label split method. However, in real-world scenarios, labels often come from different sources. For instance, in biological PPI networks, a protein may not have all its functions or phenotypes collected, or different phenotypes might be annotated by different teams, meaning not every node in a batch has all labels. Thus, the second setting, label split, was introduced: each label is assigned to the same number of nodes. Since multi-label classification on graph data is inherently transductive—where all node features and the graph structure are known—there is no risk of data leakage. This setting better reflects real-world scenarios. Note that split setting is not split ratio setting which means different proportions of training samples. Please refer to Tab. 1 for node split setting. Here in appendix, we show the performance under label split setting in Tab. 6 and Tab. 7.

**Results.** In our experiments, we employed two different split ratios, specifically, $6 : 2 : 2$ and $2 : 2 : 6$. Under the same split ratio, we repeat the random splitting process with different random seeds five times.

Due to its abundance of labels, better reveals whether the model truly uncovers the rich relationships among labels. From the Tab. 6, we can draw several conclusions. First and foremost, our method is the most effective one most of the time, which verifies the effectiveness of our approach.

Table 6: Performance under *label split* setting with the split of 6:2:2 (mean $\pm$ std. deviation) in terms of macro F1 , macro ROC_AUC. The top performance are bolded and second ones are underlined.

| Metrics | DBLP | | Delve-M | | BlogCat | | Abnomal | |
|---|---|---|---|---|---|---|---|---|
| | Macro F1 | Macro AUC | Macro F1 | Macro AUC | Macro F1 | Macro AUC | Macro F1 | Macro AUC |
| Node2vec+BR | 75.92±0.34 | 83.78±0.48 | 50.34±1.11 | 64.03±1.49 | 52.01±1.41 | 60.08±0.97 | 43.55±1.88 | 76.59±1.17 |
| Node2vec+CC | 73.61±0.31 | 80.34±0.32 | 50.66±1.29 | 65.47±1.44 | 52.22±1.39 | 60.59±1.58 | 45.12±.143 | 79.28±1.44 |
| GCN+ML-KNN | 77.64±0.35 | 86.62±0.41 | 51.98±1.24 | 69.45±1.81 | 53.61±1.14 | 63.56±2.62 | 47.22±3.24 | 80.28±1.81 |
| GCN+PLAIN | 82.27±0.34 | 91.28±0.44 | 53.61±1.74 | 78.72±1.57 | 56.30±2.29 | 70.69±1.38 | 52.16±3.27 | 93.23±1.89 |
| MLGW | 79.13±0.66 | 90.14±0.63 | 52.25±0.43 | 74.15±0.18 | 52.97±1.20 | 68.70±1.14 | 49.12±2.27 | 82.14±1.92 |
| ML-GCN | 78.64±0.44 | 89.30±0.28 | 51.92±0.34 | 83.21±0.84 | 55.29±0.22 | 70.64±0.68 | 51.27±1.30 | 86.61±1.21 |
| LARN | 83.86±0.28 | 92.77±0.33 | 54.31±1.29 | 85.85±1.15 | 55.61±1.11 | 72.44±2.12 | 51.34±1.44 | 88.61±1.49 |
| LANC | 80.37±0.62 | 89.58±0.49 | 55.14±0.58 | 86.48±0.62 | 54.48±0.84 | 70.55±1.04 | 50.23±0.66 | 84.02±0.33 |
| VariMul | 84.81±0.67 | 93.17±0.38 | 56.81±0.23 | 86.91±0.13 | 57.88±1.34 | 71.35±0.49 | 52.32±0.50 | 88.56±0.36 |
| GCN+Com | 86.00±0.24 | **94.00±0.05** | 56.82±0.28 | 88.74±0.12 | 57.39±1.75 | 70.29±2.38 | 50.86±0.25 | 87.08±0.26 |
| GCN+Auto | 85.43±0.10 | 93.47±0.14 | 54.71±0.29 | 79.08±0.17 | 53.44±1.50 | 66.41±2.37 | 52.35±0.23 | 90.31±0.31 |
| GCN+**LIP** | **86.50±0.34** | 93.91±0.19 | **58.87±1.26** | **90.89±1.14** | **59.02±0.21** | **74.30±0.36** | **53.98±0.34** | **94.91±0.45** |

Here we show the second setting of training label ratio in Tab. 7. We simulated a scenario with sparse training samples, as often seen in real-world applications. In this setting, the limited training labels make it more challenging to capture label relationships. Methods that introduce labels as new nodes and construct a new graph with label-node edges are more affected, as fewer training labels result in fewer connections between the new label nodes and the original nodes, potentially leading to poorer performance.

Table 7: Performance under *label split* setting with the split of 2:2:6 (mean $\pm$ std. deviation) in terms of macro F1, macro ROC_AUC. The top performance are bolded and second ones are underlined.

| Metrics | DBLP | | Delve-M | | BlogCat | | Abnomal | |
|---|---|---|---|---|---|---|---|---|
| | Macro F1 | Macro AUC | Macro F1 | Macro AUC | Macro F1 | Macro AUC | Macro F1 | Macro AUC |
| Node2vec+BR | 72.02±1.15 | 77.76±1.14 | 40.19±0.82 | 53.42±1.54 | 48.33±0.37 | 54.08±1.41 | 45.51±1.82 | 71.32±2.12 |
| Node2vec+CC | 71.52±2.14 | 75.59±1.59 | 42.45±1.03 | 56.56±1.44 | 49.57±1.51 | 55.84±1.00 | 45.41±2.88 | 71.33±2.32 |
| GCN+ML-KNN | 74.43±2.12 | 79.59±2.18 | 45.34±1.76 | 57.49±2.29 | 51.08±1.82 | 59.24±1.22 | 45.81±1.72 | 73.39±0.33 |
| GCN+PLAIN | 79.34±1.32 | 88.67±0.52 | 49.23±1.25 | 61.25±1.54 | 55.34±1.26 | 67.35±2.11 | 48.23±1.32 | 87.11±2.42 |
| MLGW | 78.62±2.27 | 85.95±1.19 | 47.24±2.56 | 57.83±2.13 | 51.82±0.36 | 60.48±0.24 | 45.68±1.15 | 74.35±1.29 |
| ML-GCN | 78.62±1.45 | 83.63±1.21 | 49.37±1.12 | 58.00±1.18 | 53.55±1.03 | 61.77±1.29 | 47.57±1.44 | 77.24±1.18 |
| LARN | 81.45±0.74 | **92.47±1.83** | 50.71±1.98 | 62.08±1.86 | 54.11±1.66 | 63.56±1.62 | 49.35±1.22 | 80.22±1.57 |
| LANC | 79.63±0.91 | 88.43±0.72 | 49.80±1.12 | 57.85±3.18 | 54.18±1.82 | 62.18±1.87 | 49.24±0.88 | 83.52±1.41 |
| VariMul | 83.23±0.43 | 91.06±1.18 | 50.39±1.18 | 59.08±1.27 | 57.02±1.23 | 69.31±1.18 | 50.54±1.05 | 88.51±2.15 |
| GCN+Auto | 83.96±0.13 | 91.26±0.16 | 47.17±0.32 | 64.44±2.69 | 56.34±1.27 | 62.64±1.01 | 48.14±1.04 | 85.12±1.51 |
| GCN+**LIP** | **84.05±0.19** | 92.21±0.28 | **52.56±1.21** | **69.33±1.34** | **57.07±0.43** | **72.40±0.66** | **51.91±1.18** | **90.58±0.52** |

As shown in Tab. 7, our method once again achieved the best performance, with greater improvement over the baseline compared to scenarios with richer training samples.

### E.3 CHANGING BACKBONE OF **LIP**

As shown in Fig. 5(a), our model consistantly achieves improvement when applied to any backbone, demonstrating the effectiveness of our approach. The difference in performance among backbones not using our method depends entirely on the inductive bias inherent in the graph data itself. An interesting observation is that different backbone models exhibit varying degrees of sensitivity to our method. For instance, GCN show greater improvements before and after the application of our method compared to GAT. We think that this may be due to our method's relation to gradient calculation, with different models having varying sensitivities to labels. Possibly, during training, GAT's gradient changes more rapidly, leading to quicker changes in label correlation. Moreover, since our method can be flexibly applied to any backbone capable of obtaining node embeddings, it allows our approach to serve as a plug-and-play enhancement for MLNC.

Moreover, we further conduct a series of experiments by replacing the backbone with more advanced decoupled GNNs: APPNP Gasteiger et al. (2019) and GPRGNN Chien et al. (2020). APPNP

Table 8: Changing backbones (AUC) under node split on MLNC datasets.

| AUC | DBLP | BlogCat | PCG | EukLoc |
|---|---|---|---|---|
| GCN | $92.83 \pm 1.13$ | $66.14 \pm 1.74$ | $59.54 \pm 0.90$ | $70.53 \pm 1.97$ |
| GCN+LIP | $94.38 \pm 1.51$ | $70.21 \pm 2.02$ | $65.73 \pm 0.52$ | $72.92 \pm 1.82$ |
| GIN | $93.00 \pm 0.46$ | $68.32 \pm 0.67$ | $63.44 \pm 1.15$ | $73.13 \pm 1.24$ |
| GIN+LIP | $94.75 \pm 1.29$ | $70.87 \pm 0.93$ | $66.10 \pm 1.64$ | $75.10 \pm 1.29$ |
| APPNP | $94.17 \pm 0.92$ | $70.33 \pm 1.10$ | $64.96 \pm 1.33$ | $74.67 \pm 0.98$ |
| APPNP+LIP | $95.21 \pm 1.08$ | $71.82 \pm 1.45$ | $65.51 \pm 1.74$ | $75.86 \pm 1.02$ |
| GPRGNN | $93.09 \pm 1.12$ | $68.31 \pm 1.26$ | $68.02 \pm 1.17$ | $72.91 \pm 0.99$ |
| GPRGNN+LIP | $95.07 \pm 1.84$ | $72.36 \pm 0.97$ | $68.74 \pm 1.58$ | $74.88 \pm 1.06$ |

is a classical yet powerful model, while GPRGNN achieve state-of-the-art (SOTA) on both homophily and heterophily graphs. Here, we evaluate the effectiveness of our method, LIP, on these advanced decoupled GNNs and present a comparison of the results between the original models and the LIP-enhanced models on MLNC tasks in the Tab. 8.

From the Tab. 8, we can observe that even for high-performance decoupled GNN models, LIP consistently achieves performance improvements. LIP provides enhancements across different domains and demonstrates exceptional results on certain datasets. For example, LIP improved the performance of GPRGNN by approximately 4% on the BlogCat dataset. In many cases, the performance improvement brought by LIP exceeds the performance differences between different models, highlighting the value of incorporating LIP. This is because these backbones primarily focus on modeling the input graph structure and node features. Regardless of the type of GNNs used, our LIP can provide additional support by modeling the influence correlation between labels.

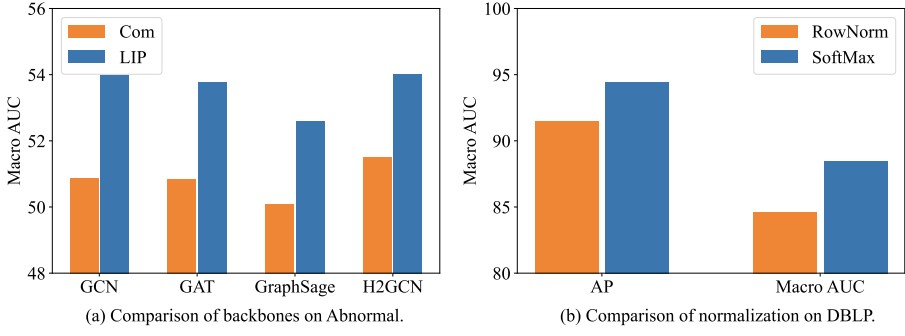

(a) Comparison of backbones on Abnormal.

(b) Comparison of normalization on DBLP.

Figure 5: More model analysis on different datasets.

### E.4    ABLATION STUDY

**Normalization.** As shown in Fig. 5(b), the softmax normalization we use is more effective. From our empirical analysis, Softmax offers two key advantages compared to Row normalization. First, Softmax provides stronger differentiation. In the case of the influence matrix, failing to capture differences in influence may result in similar learning weights for labels that either negatively impact or provide only minimal positive impact, potentially leading to a vicious cycle. Second, Softmax provides stable gradients, contributing to smoother model optimization. In contrast, Row normalization, being a simple normalization operation, may gradually flatten the differences in influence relationships over successive iterations.

**Influence Correlations in P and T steps.** We supplement the ablation study across all datasets (the full version of Fig. 4b), which is shown in the Tab. 9. "None" stands for simply using the backbone model without any quantification of influence correlations between labels.

It can be observed that in all cases, utilizing the influence correlations from both propagation (P) and transformation (T) steps (noted as All in the table) achieves the best performance than using the influence from either phase alone. Moreover, individually quantifying and utilizing either type of influence correlations yields better performance than not using them at all. This indicates that

Table 9: Ablation study of label influences (GCN as backbone).

| AUC | DBLP | BlogCat | OGB-p | PCG | HumLoc | EukLoc |
|---|---|---|---|---|---|---|
| None | $92.83 \pm 1.13$ | $66.14 \pm 1.74$ | $71.26 \pm 1.45$ | $59.54 \pm 0.90$ | $66.57 \pm 0.67$ | $69.27 \pm 1.97$ |
| Only P | $92.08 \pm 1.06$ | $68.27 \pm 1.88$ | $73.72 \pm 0.63$ | $62.01 \pm 1.21$ | $70.17 \pm 1.42$ | $70.87 \pm 1.64$ |
| Only T | $93.94 \pm 1.00$ | $67.11 \pm 1.51$ | $73.58 \pm 1.24$ | $63.82 \pm 1.04$ | $69.30 \pm 1.02$ | $69.93 \pm 1.01$ |
| All | $\mathbf{94.38 \pm 1.51}$ | $\mathbf{70.21 \pm 2.02}$ | $\mathbf{74.82 \pm 0.34}$ | $\mathbf{65.73 \pm 0.52}$ | $\mathbf{73.22 \pm 1.76}$ | $\mathbf{72.92 \pm 1.82}$ |

leveraging the influence relationships from both the P and T steps is crucial for the MLNC task. Additionally, the table reveals that using the influence correlations from either the P step or the T step alone can achieve better results on different datasets. We hypothesize that this is due to the varying demands of different datasets for the P and T processes. Some datasets may require minimizing negative influence during the P process, while others may benefit from maximizing positive influence during the T process.

### E.5 SENSITIVITY STUDY

**Settings.** We examine the sensitivity of the model to hyper-parameters by varying the values of $\alpha$ and $\beta$. We draw inspiration from APPNP's (Gasteiger et al., 2019) experimental setup. When we vary $\alpha$, $\beta$ is fixed at 0.28; when we vary $\beta$, $\alpha$ is fixed at 0.15.

**Results.** The conclusion is that our method demonstrates robustness across a range of hyper-parameter settings. Specifically, the performance remains stable within reasonable parameter ranges, indicating that the model does not heavily depend on fine-tuned hyper-parameters for achieving effective results. Moreover, we find that regardless of the dataset characteristics, the optimal range for $\alpha$ is approximately between 0.08 and 0.30, while the optimal range for $\beta$ is approximately between 0.10 and 0.56.

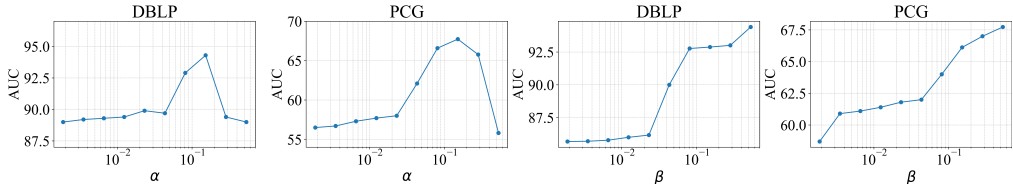

Figure 6: Changing hyper-parameters of $\alpha$ and $\beta$ on datasets DBLP and PCG.

### E.6 EVALUATION UNDER INDUCTIVE SETTING

**Settings.** In this section, we perform the inductive train/val/test split (6:2:2) and conducted experiments to validate the effectiveness in such setting. We used GraphSage as the backbone because it is naturally suited for the inductive setting. We also excluded certain baselines that are not suitable for the setting, such as VariMul and MLGW.

**Results.** The results are shown in the Tab. 10. It shows that our method also achieves satisfactory performance under the inductive setting. Although the model cannot observe the complete graph in the inductive setting, the subgraph containing the nodes whose labels need to be predicted is visible. Therefore, our model's quantification of influence correlation during the P step remains meaningful and effective. Since inductive training also involves mutual interactions between gradients of different labels, modeling the influence relationships between labels in the T step is also necessary. As a result, our method achieves performance surpassing the baselines even under the inductive setting.

### E.7 CHANGING NUMBER OF LABELS

**Settings.** We used GCN as the backbone to compare and analyze the performance of our method across different label number of label categories $k$.

Table 10: Performance (AUC) comparison under inductive setting.

|  | SAGE+ML-KNN | SAGE+PLAIN | LARN | LANC | SAGE+Auto | SAGE+LIP |
|---|---|---|---|---|---|---|
| DBLP | $72.45 \pm 1.77$ | $74.16 \pm 0.91$ | $73.87 \pm 1.79$ | $73.54 \pm 1.95$ | $77.11 \pm 2.42$ | **$79.32 \pm 1.96$** |
| EukLoc | $53.31 \pm 1.51$ | $55.02 \pm 1.73$ | $63.39 \pm 2.01$ | $65.97 \pm 1.67$ | $62.82 \pm 2.19$ | **$66.35 \pm 2.36$** |

**Results.** From the Tab. 11, we can observe that our method consistently achieves performance boosting regardless of the number of labels. We also identified some interesting phenomena from the experiments. First, for GCN, the performance is actually the best when the number of labels is 2. This phenomenon can be understood from two perspectives: On one hand, statistical analysis shows that label 2 is the category with the highest number of training nodes across all training datasets. Therefore, when the total number of labels is 2, the contribution of label 2 leads to a higher average performance. On the other hand, this also indicates that, to some extent, an increase in the number of labels results in a decline in GCN's prediction performance.

Moreover, with our method, the performance generally improves as the number of labels increases. In contrast, GCN's performance fluctuates as the number of labels increases, sometimes improving and sometimes deteriorating. This indirectly indicates that our method reduces the negative influence between labels while enhancing the positive influence.

Table 11: Performance (AUC) when changing the number of label categories $k$ on HumLoc.

| HumLoc (AUC) | 2 | 4 | 6 | 8 | 10 | 12 | 14 |
|---|---|---|---|---|---|---|---|
| GCN | $70.29 \pm 1.74$ | $64.44 \pm 0.92$ | $66.35 \pm 1.49$ | $65.14 \pm 1.85$ | $66.52 \pm 1.22$ | $66.73 \pm 0.86$ | $68.14 \pm 1.88$ |
| GCN+LIP | $72.17 \pm 1.56$ | $65.99 \pm 1.02$ | $68.41 \pm 1.55$ | $68.82 \pm 0.97$ | $72.13 \pm 1.29$ | $72.22 \pm 1.76$ | $73.22 \pm 1.76$ |

### E.8 CASE STUDY

**Settings.** To better illustrate the positive and negative mutual influences between labels, we take a specific node and its surrounding neighbors as an example to demonstrate the mutual influences between labels. We aim to show how these influences contribute to either improving or impairing the prediction performance of the node. We selected a node with a degree of 3 from the DBLP dataset, identified as node 197. Its adjacency structure in the original graph is shown in the Fig. 7.

**Analysis.** It is worth noting that, as discussed in Sec. 4 of the main text, the influence between labels is quite complex. The influence from $Y_a$ to $Y_b$ is the combined effect of a group of nodes with label $y_a$ on a group of nodes with label $y_b$ during both the P and T processes. Here, we attempt to visualize the positive and negative influences, as well as their ultimate effect on performance, using the local structure of a single node rather than a group of nodes with the same label. This provides a simplified perspective on these interactions.

From the local structure of node 197, we can observe its connections to nodes 0, 18566, and 22, along with their respective node indices and ground truth labels, as shown in the figure. However, when GCN predicts the labels for node 197, the resulting predicted probabilities are shown in the bar chart. Based on the observations from Fig. 1(a), we can roughly infer the reasons for this phenomenon:

Positive and Negative Influences: From Fig. 1(a), it is evident that the other three labels generally provide positive support for label 0 and label 2, while they mostly exert negative influence on label 1 and label 3. This implies that the influence from other labels makes it easier for node 197 to be correctly identified as label 0 or label 2, but not necessarily for label 1 or label 3.

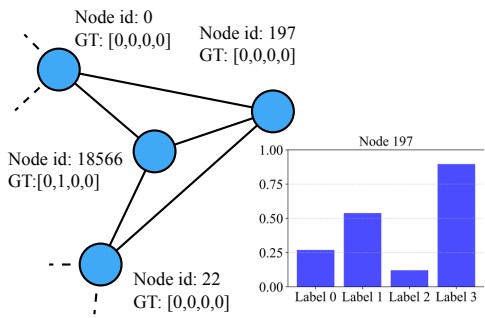

Figure 7: Case study on node 197 from dataset DBLP.

Localized Negative Influence: Among the neighboring nodes of node 197, only node 18566 has label 1, which is different from its surroundings. Consequently, its negative influence is received by node 197 during the P process, resulting in label 1's probability exceeding 0.5.

Impact on label 3: The high predicted probability for label 3 is likely due to the influence of other labels during backpropagation, which affects the GNN parameters. This, in turn, increases the probability of label 3 during the T process. In conclusion, the combined effects of positive and negative influences ultimately impact the final prediction. As a result, Labels 0 and 2 are correctly predicted for node 197, while Labels 1 and 3 are incorrectly predicted.

## F    LIMITATIONS AND FUTURE WORK

From a scenario perspective, our work currently focuses on the most common case: static and homogeneous graphs. That is, the nodes, edges, and labels do not change over time, and all nodes in the dataset are of the same type, with all edges sharing the same physical meaning. From a goal perspective, our work is currently focused on improving the performance of MLNC tasks by analyzing and quantifying the influence correlations between labels on graph datasets, without considering potential noise in the labels or graph structure. We leave these complex challenges in both aspects for future exploration.

