# OpenReview forum: "Multi-Label Node Classification with Label Influence Propagation"
_ICLR.cc/2025/Conference — ICLR 2025 Poster_

### Official Review · Reviewer_SyKm · 2024-10-23

**Soundness:** 3
**Presentation:** 2
**Contribution:** 3
**Rating:** 6
**Confidence:** 3

**Summary:**

This paper proposes the Label Influence Propagation (LIP) method for multi-label node classification on graph-structured data. The main idea is to model both positive and negative influences between labels by separating the message-passing process into distinct propagation and transformation operations. Additionally, LIP constructs a label influence graph to quantify label-wise importance scores.

**Strengths:**

- This paper is well-motivated, emphasizing the importance of multi-label node classification in various domains. The challenge of label correlations and their potential positive and negative influences in non-Euclidean data is clearly explained.
- The idea of constructing a label graph is interesting.
- Experimental results demonstrate that LIP achieves notable performance gains across datasets of diverse domains, regardless of the backbone GNNs, highlighting its versatility.

**Weaknesses:**

**W1.** The writing needs more thorough proofreading. I noticed several grammatical issues, which detract from the overall quality of the paper.

Examples include:
- "Illustrate" should be changed to "Illustration of" in Fig. 3 caption.
- "contain" should be "containing" in line 300.
- "analysis" should be "analyze" in line 313.
- "which can be change" should be "which can be changed" in line 320.

Additionally, table captions should be placed *above* the tables.

Several critical errors related to definitions also needs to be revised:
- "negative influence" should be "positive influence" in line 301.
- The eq. 6 seems inconsistent with the textual explanation of positive influence.
- $\Omega_j$ may need to be revised to $\psi_j$ for consistency.

**W2.** The clarity of the paper needs to be improved. For instance:
- The theoretical justification in Section 4.2 and Appendix A needs more clarity. While the authors assert that the graph structure is a key driver of label influence during propagation, they do not fully clarify how the feature influence $I(i,j)$ and PPR are connected to the proposed label influence metric in the propagation operation. I can infer that positive and negative influences in PPR and feature influence metrics correspond to Equations 6 and 5, respectively, but this connection should be made explicit.
- Additionally, the augmented form of $\text{\textbf A}$ is not clearly defined. Is it a multi-hop adjacency matrix?

**W3.** What are the limitations of the proposed method? The authors didn't include a discussion on the potential limitations of the proposed method.

If the above concerns and subsequent questions are addressed, I'm willing to raise my score.

**Questions:**

**Q1.** Could you provide additional explanation of the performance enhancement on heterophilous graphs? It's interesting that LIP consistently enhances performance on these datasets, despite the method appearing to be built upon a homophily assumption.

**Q2.** How does the performance change when varying the $\beta$ in eq. 12?

**Q3.** Although the authors state that "multi-label classification on graph data is inherently transductive" in the Appendix, the inductive setting with partially accessible graph structure is more realistic in many real-world applications. While benchmark datasets are commonly used in a transductive manner, it would be straightforward to modify these datasets for the inductive setting by masking nodes and their corresponding edges in different splits. The authors should consider evaluating LIP under such conditions to verify the practical relevance.

---

> ### Author Response · Authors · 2024-11-21
> **Response to Reviewer SyKm (1/3)**
>
> Dear reviewer SyKm,
> Thank you for your insightful comments. Please, see below our answer to the raised comments/questions.
>
> > **W1. Several grammatical issues need proofreading.**
>
> Thank you very much for your thorough review and detailed suggestions.
>
> We have made the corresponding revisions in the revised pdf and highlighted the revisions in blue.
> We will continue to polish the paper to further enhance its readability.
> Moreover, we rewrite the explanation preceding Def. 4 to provide a detailed explanation of the positive influence.
>
> > **W2.1 How the influence $I(i, j)$ and PPR are connected to the proposed label influence metric in the propagation operation?**
>
> We have revised the motivation and analysis in Sec 4.2 and Appendix A to better clarify the relationship between the influence among labels during the propagation (P) step and $I(i, j)$, PPR.
>
> Overall, we decompose the computation of influence correlations among labels during the P step into ***two parts***: the magnitude of influence between all pairs of nodes ("Influence between node pairs" at line 247) and the positive or negative direction of influence between node pairs with different labels ("Influence between label sets" at line 268).
>
> For ***the first part***, our analysis shows that the magnitude of influence between any pair of nodes during the P phase can be defined as $I(i, j)$.
> Although the calculation of $I(i, j)$ initially involves node features, inspired by [1], we prove that the magnitude of $I(i, j)$ is actually independent of the features themselves and is instead proportional in expectation to a random walk distribution $\mathbf{\pi}\_{\text{lim}}^{(ji)}$, namely $I(i, j) \propto \mathbf{\pi}\_{\text{lim}}^{(ji)}$.
> To further quantify and compute this distribution $\mathbf{\pi}\_{\text{lim}}$, inspired by [2], we find that we can solving the PPR $\mathbf{\pi}\_{\text{ppr}}$ instead.
> By iteratively computing the PPR, we can obtain the influence correlations between any two nodes.
> Thus, we derive the quantification of the influence between any pair of nodes $v_i, v_j$ during the P phase as: $\mathbf{INF}^{P}(v_i, v_j) := I(i, j) \propto \mathbf{\pi}\_{\text{ppr}}^{(ji)} = \{ \alpha (\mathbf{I}\_n - (1 - \alpha)\hat{\mathbf{A}})^{-1} \mathbf{s}_{v_i} \}\_{v_j}$
>
> Having computed the influence correlations magnitude $\mathbf{INF}^{P}(v_i, v_j)$ between any pair of nodes during the P phase, represented as an  $n\times n$ node influence correlation matrix, ***the second part*** involves integrating the influence from all nodes in  $Y_a$  (with label $y_a$) on all nodes in  $Y_b$ (with label $y_b$) to get the label influence correlation between $y_a$ and  $y_b$. This part yields a $k \times k$ label influence correlation matrix $\mathbf{INF}^{P}(y_a,y_b)$.
> Through our analysis, we find that the influence from $Y_a$ to $Y_b$ has both positive and negative directions. Based on the analysis in Fig. 3, we identified the node sets contributing to the positive (Eq. 6) and negative influences (Eq. 5)  for any pair of labels. By incorporating these signs and integrating the pairwise influences between nodes $\mathbf{INF}^{P}(v_i, v_j)$, we can ultimately compute the influence correlations between labels $\mathbf{INF}^{P}(y_a,y_b)$.
>
> In short, the label influence correlation $\mathbf{INF}^{P}$ in the P operation is defined as $I(i, j)$, which can be directly computed using PPR.
>
> > **W2.2 What is the augmented form of $A$?**
>
> We have added a more detailed description in line 252 of revised version. Specifically, the augmented form of $\mathbf{A}$ that we use is $\hat{\mathbf{A}} = \mathbf{{\tilde{D}}}^{-1/2} \mathbf{{\tilde{A}}} \mathbf{{\tilde{D}}}^{-1/2}$ [2].
> Namely, $\hat{\mathbf{A}}$ is the symmetrically normalized adjacency matrix with self-loops, where $\mathbf{\tilde{A}}= \mathbf{A} + \mathbf{I}\_n$ is the adjacency matrix with added self-loops, $\mathbf{{\tilde{D}}}\_{ij} = \delta\_{ij} \sum\_k \mathbf{{\tilde{A}}}\_{ik}$ is the diagonal degree matrix [3], $\delta_{ij}$ is the Kronecker delta function indicating the edge between $i$ and $j$.
>
> > **W3. What are the limitations of the method?**
>
> We have added a paragraph of "limitation and future work" in the Appendix.
>
> From a scenario perspective, our work currently focuses on the most common case: static and homogeneous graphs. That is, the nodes, edges, and labels do not change over time, and all nodes in the dataset are of the same type, with all edges sharing the same physical meaning.
> From a goal perspective, our work is currently focused on improving the performance of MLNC tasks analyzing and quantifying the influence correlations between labels on graph datasets, without considering potential noise in the labels or graph structure.
> We leave these complex challenges in both aspects for future exploration.

---

> ### Author Response · Authors · 2024-11-21
> **Response to Reviewer SyKm (2/3)**
>
> > **Q1. Provide additional explanation of the performance enhancement on heterophilous graphs.**
>
> Sorry for the misleading expressions. We have polished the paper accordingly.
>
> *Actually, our analysis is based on the Laplacian smoothing effect inherent in the propagation (P) step rather than homophily*.
>
> That is, as the number of propagation or aggregation layers in the message passing increases, the representations and predicted labels of neighboring nodes tend to become increasingly similar.
> In simple terms, the negative influence during the P step, originates from nodes in $Y\_a$ and $Y\_b$ (excluding their intersection) that are supposed to have different labels but become similar during the P step. Conversely, the positive influence comes from nodes with the same labels, such as nodes in $Y\_b$ without the intersection, which become more similar during the P process, making it easier to achieve correct label predictions.
>
> *In fact, the design of our method is independent of whether the graph follows the homophily assumption.*
>
> As long as the backbone model includes a propagation or aggregation step, the influence during these processes can be analyzed as described above (more detailed in Sec 4.2). We mentioned homophily mainly because, as noted in [1], the smoothing of propagation step significantly aids homophily graphs. However, homophily is not a necessary condition for the P step. Existing graph models specifically designed for heterophily graphs also include propagation steps [6].
>
> Therefore, our conclusion that the method achieves performance improvement on both homophily and heterophily graphs is based on the following:
> 1. Our quantification of label influence correlations during the P step is grounded in the Laplacian smoothing, which is present in the P step from the models designed for both homophily and heterophily graphs.
> 2. We also quantify the label influence correlations during the T step and construct a label influence graph to capture higher-order influence correlations. This allows us to enhance/suppress the positive/negative label influences, which is independent of whether the graph is homophilic or heterophilic.
> 3. Our method is designed independent of the backbone model. For heterophily graphs, we can use graph models specifically designed for heterophily graphs as backbone to achieve further performance improvements.
>
>
> > **Q2. How does the performance change when varying the $\beta$?**
>
> We have added a hyper-parameter sensitivity study in Appendix E.5, especially $\beta$.
>
> The conclusion is that our method demonstrates robustness across a range of hyper-parameter settings. Specifically, the performance remains stable within reasonable parameter ranges, indicating that the model does not heavily depend on fine-tuned hyper-parameters for achieving effective results.
> Moreover, we find that regardless of the dataset characteristics, the optimal range for $\beta$ is approximately between 0.10 and 0.56.

---

> ### Author Response · Authors · 2024-11-21
> **Response to Reviewer SyKm (3/3)**
>
> > **Q3. How is the performance under specific inductive setting?**
>
> Thank you for your practical suggestions regarding the inductive setting.
>
> Following your advice, we perform the inductive train/val/test split (6:2:2) and conducted experiments, which we have added in Appendix E.6.
> Due to time constraints, we have supplemented the results with comparisons on two datasets from different domains. We used GraphSage as the backbone because it is naturally suited for the inductive setting. We also excluded certain baselines that are not suitable for the setting, such as VariMul and MLGW.
> The results are shown in the table below.
>
> Table 1. Performance (AUC) comparison under inductive setting.
>
> |        |   SAGE+ML-KNN    |    SAGE+PLAIN    |       LARN       |       LANC       |    SAGE+Auto     |       SAGE+LIP       |
> | :----: | :--------------: | :--------------: | :--------------: | :--------------: | :--------------: | :------------------: |
> | DBLP  | 72.45 $\pm$ 1.77 | 74.16 $\pm$ 0.91 | 73.87 $\pm$ 1.79 | 73.54 $\pm$ 1.95 | 77.11 $\pm$ 2.42 | **79.32 $\pm$ 1.96** |
> | EukLoc | 53.31 $\pm$ 1.51 | 55.02 $\pm$ 1.73 | 63.39 $\pm$ 2.01 | 65.97 $\pm$ 1.67 | 62.82 $\pm$ 2.19 | **66.35 $\pm$ 2.36** |
>
>  It shows that our method also achieves satisfactory performance under the inductive setting. Although the model cannot observe the complete graph in the inductive setting, the subgraph containing the nodes whose labels need to be predicted is visible. Therefore, our model’s quantification of influence correlations during P step remains meaningful and effective. Since inductive training also involves mutual interactions between gradients of different labels, modeling the influence relationships between labels in the T step is also necessary. As a result, our method achieves performance surpassing the baselines even under the inductive setting.
>
>
> [1] Representation Learning on Graphs with Jumping Knowledge Networks (JKNet).
>
> [2] Predict then propagate: Graph neural networks meet personalized pagerank (APPNP).
>
> [3] Semi-supervised classification with graph convolutional networks (GCN).
>
> [4] Model Degradation Hinders Deep Graph Neural Networks.
>
> [5] Deeper insights into graph convolutional networks for semi-supervised learning.
>
> [6] The Heterophilic Graph Learning Handbook: Benchmarks, Models, Theoretical Analysis, Applications and Challenges.

---

> > ### Comment · Reviewer_SyKm · 2024-11-22
> > **Official Comment by Reviewer SyKm**
> >
> > Thank you for your detailed rebuttal. The authors have thoroughly addressed all of my concerns and clarified some of my misunderstandings. As a result, I have raised my score to 6.

---

> > > ### Author Response · Authors · 2024-11-22
> > >
> > > Dear reviewer SyKm,
> > >
> > > Thank you very much for your time reviewing our answer, and for updating your score.
> > > We appreciate your valuable suggestions and insightful questions.

---

### Official Review · Reviewer_SoZZ · 2024-10-25

**Soundness:** 3
**Presentation:** 3
**Contribution:** 2
**Rating:** 6
**Confidence:** 4

**Summary:**

This paper develops a new method LIP that leverages the propagation strategy to obtain high-order label information in the graph for multi-label node classification . The authors provide the theoretical analysis for the motivation and the proposed method. The extensive experimental results show the effectiveness of LIP on multi-label node classification.

**Strengths:**

1.  This paper is well-written and easy to follow.
2.  The authors provide the theoretical guarantee for the proposed method.
3.  LIP shows promising performance on various datasets.

**Weaknesses:**

1.  The review of related work is not comprehensive.
2.  The ablation study is inadequate.
3.  The efficiency study is missing.

**Questions:**

My concerns are mainly from two parts: discussions of relation work and designs of experiments.

1.  Actually, decoupled GNNs\[1,2,3,4,5] have been studied in past a few years. Although the authors are inspired by the theoretical analysis to decouple the propagation module and feature transformation, the previous efforts in decoupled GNNs should be discussed. I have noticed the authors cite APPNP \[1], one of the representative decoupled GNNs, in Line 263. Here, I suggest the authors open a new subsection in Related Work to comprehensively review recent decoupled GNNs.
2.  As a plug-and-play, I suggest the authors try more advanced GNNs as backbones for ablation study, such as advanced decoupled GNNs\[1,3].
3.  Based on Figure 4(b), I suggest the authors conduct ablation on all other datasets to comprehensively validate the contribution of each module.
4.  The author claim the efficiency of the proposed method via the time complexity analysis. Maybe the authors can report the computational cost of each methods, including training time cost and GPU memory cost, to strength this contribution.



[1] Predict then propagate: Graph neural networks meet personalized pagerank, ICLR 2019

[2] On the equivalence of decoupled graph convolution network and label propagation, WWW 2021

[3] Adaptive universal generalized pagerank graph neural network, ICLR 2021

[4] Towards deeper graph neural networks, KDD 2020

[5] Neighborhood Convolutional Graph Neural Network, KBS 2024

---

> ### Author Response · Authors · 2024-11-21
> **Response to Reviewer SoZZ (1/2)**
>
> Dear reviewer SoZZ,
> Thank you for your insightful comments. Please, see below our answer to the raised comments/questions.
>
> > **Q1. Discussion about decoupled GNNs.**
>
> Thank you for your valuable suggestions.
> We have added a paragraph in Sec. 2 of the paper as suggested. For a detailed discussion on decoupled GNN-related works, please refer to the additional paragraph in the revised paper. Here, we focus on discussing the relationship between our work and decoupled GNNs.
>
> Our work is inspired by these studies that disentangle [1] or decouple [2,3,4] the message passing process into propagation (P) and transformation (T) operations.  We focus more on analyzing the influence correlations between labels during the P and T operations, rather than designing specific P and T operations to increase model performance [2,3] or enhance model capacity and scalability [4].
> Moreover, as shown in the backbone replacement experiments in the response to the next question, our method achieves performance improvement regardless of whether the backbone is a decoupled or non-decoupled GNN.
>
> > **Q2. Try more advanced decoupled GNNs as backbones.**
>
> We have conducted a series of experiments by replacing the backbone with more advanced decoupled GNNs: APPNP [2] and GPRGNN [3]. APPNP is a classical yet powerful model, while GPRGNN achieve state-of-the-art (SOTA) on both homophily and heterophily graphs.
> We have also added experiments with other alternative backbones in Appendix E.4 of the revised pdf.
> Here, we evaluate the effectiveness of our method, LIP, on these advanced decoupled GNNs and present a comparison of the results between the original models and the LIP-enhanced models on MLNC tasks in the table below.
>
> Table 1. Changing backbones (AUC) under node split on MLNC datasets.
>
> |    AUC     |       DBLP       |     BlogCat      |       PCG        |      EukLoc      |
> | :--------: | :--------------: | :--------------: | :--------------: | :--------------: |
> |    GCN     | 92.83 $\pm$ 1.13 | 66.14 $\pm$ 1.74 | 59.54 $\pm$ 0.90 | 70.53 $\pm$ 1.97 |
> | GCN+LIP   | 94.38 $\pm$ 1.51 | 70.21 $\pm$ 2.02 | 67.73 $\pm$ 0.52 | 74.92 $\pm$ 1.82 |
> |   APPNP    | 94.17 $\pm$ 0.92 | 70.33 $\pm$ 1.10 | 64.96 $\pm$ 1.33 | 74.67 $\pm$ 0.98 |
> | APPNP+LIP | 95.21 $\pm$ 1.08 | 71.82 $\pm$ 1.45 | 67.51 $\pm$ 1.74 | 75.86 $\pm$ 1.02 |
> |   GPRGNN   | 93.09 $\pm$ 1.12 | 68.31 $\pm$ 1.26 | 68.02 $\pm$ 1.17 | 72.91 $\pm$ 0.99 |
> | GPRGNN+LIP | 95.07 $\pm$ 1.84 | 72.36 $\pm$ 0.97 | 68.74 $\pm$ 1.58 | 74.88 $\pm$ 1.06 |
>
> From the table above, we can observe that even for high-performance decoupled GNN models, LIP consistently achieves performance improvements. LIP provides enhancements across different domains and demonstrates exceptional results on certain datasets. For example, LIP improved the performance of GPRGNN by approximately 4% on the BlogCat dataset.
> In many cases, the performance improvement brought by LIP exceeds the performance differences between different models, highlighting the value of incorporating LIP.
> This is because these backbones primarily focus on modeling the input graph structure and node features. Regardless of the type of GNNs used, our LIP can provide additional support by modeling the influence correlation between labels.
>
>
> [1] Model Degradation Hinders Deep Graph Neural Networks.
>
> [2] Predict then propagate: Graph neural networks meet personalized pagerank (APPNP).
>
> [3] Adaptive universal generalized pagerank graph neural network (GPRGNN).
>
> [4] Neighborhood Convolutional Graph Neural Network (NCGNN).

---

> ### Author Response · Authors · 2024-11-21
> **Response to Reviewer SoZZ (2/2)**
>
> > **Q3. Conduct ablation study of Fig. 4b on all other datasets.**
>
> We supplement the ablation study across all datasets (the full version of Fig. 4b), which is shown in the table below.
> "None" stands for simply using the backbone model without any quantification of influence correlations between labels.
>
> Table 2. Ablation study of label influences on all datasets (GCN as backbone).
>
> | AUC    | DBLP                 | BlogCat              | OGB-p                | PCG                  | HumLoc               | EukLoc               |
> | ------ | -------------------- | -------------------- | -------------------- | -------------------- | -------------------- | -------------------- |
> | None   | 92.83 $\pm$ 1.13     | 66.14 $\pm$ 1.74     | 71.26 $\pm$ 1.45     | 59.54 $\pm$ 0.90     | 66.57 $\pm$ 0.67     | 69.27 $\pm$ 1.97     |
> | Only P | 92.08 $\pm$ 1.06     | 68.27 $\pm$ 1.88     | 73.72 $\pm$ 0.63     | 62.01 $\pm$ 1.21     | 70.17 $\pm$ 1.42     | 71.87 $\pm$ 1.64     |
> | Only T | 93.94 $\pm$ 1.00     | 67.11 $\pm$ 1.51     | 73.58 $\pm$ 1.24     | 65.82 $\pm$ 1.04     | 69.30 $\pm$ 1.02     | 69.93 $\pm$ 1.01     |
> | All    | **94.38 $\pm$ 1.51** | **70.21 $\pm$ 2.02** | **74.82 $\pm$ 0.34** | **67.73 $\pm$ 0.52** | **73.22 $\pm$ 1.76** | **74.92 $\pm$ 1.82** |
>
> It can be observed that in all cases, utilizing the influence correlations from both propagation (P) and transformation (T) steps (noted as All in the table) achieves the best performance than using the influence from either phase alone.
> Moreover, individually quantifying and utilizing either type of influence correlations yields better performance than not using them at all.
> This indicates that leveraging the influence relationships from both the P and T steps is crucial for the MLNC task.
> Additionally, the table reveals that using the influence correlations from either the P step or the T step alone can achieve better results on different datasets. We hypothesize that this is due to the varying demands of different datasets for the P and T processes. Some datasets may require minimizing negative influence during the P process, while others may benefit from maximizing positive influence during the T process.
>
>
> > **Q4.  Report the computational cost of each method, including training time and GPU memory.**
>
> We have added a table below showing the average per-epoch training time and GPU memory usage of various methods on graph data from two different domains.
>
> Table 3. Computational cost of each method, including time and space during training.
>
> |         | Cost           | ML-KNN | MLGW | LANC | VariMul | GCN+Auto | GCN+LIP |
> | ------- | -------------- | ------ | ----- | ----- | ------- | -------- | ------- |
> | DBLP    | Time (s/epoch) | 0.012 | 5.710 | 2.350 | 1.015   | 0.003    | 0.008   |
> | BlogCat | Time (s/epoch) | 0.100 | 9.710 | 3.837 | 2.082   | 0.051    | 0.311   |
> | DBLP    | GPU mem (MB) | 2200   | 3530 | 2204 | 2980    | 2271     | 1595    |
> | BlogCat | GPU mem (MB) | 3010   | 5036 | 2800 | 3572    | 3073     | 2369    |
>
>
> As observed, our method aligns with the conclusions analyzed in Appendix B, demonstrating relatively shorter training times and especially lower GPU memory usage. This indicates that our method has better scalability.
> Specifically, our training time is roughly on the same order of magnitude as a standard GCN but takes a few times bigger. However, compared to other baselines, our method typically requires one order of magnitude less time. On the other hand, since the quantification of influence correlations during the T step involves gradient calculations, our method essentially splits the gradient calculation into k smaller steps. This trades off some computation time for reduced memory usage, allowing our method to achieve even lower GPU memory consumption than a standard GCN.

---

> > ### Comment · Reviewer_SoZZ · 2024-11-22
> >
> > Thanks for the authors' efforts. I have raised my score and wish the authors can incorporate the above discussions into the final version.

---

> > > ### Author Response · Authors · 2024-11-22
> > >
> > > Dear Reviewer SoZZ,
> > >
> > > Thank you very much for your time reviewing our responses, and for raising your score.
> > > We will ensure to include these discussions in our final paper.
> > > Your valuable feedback has made our work better.

---

### Official Review · Reviewer_jPZg · 2024-11-02

**Soundness:** 3
**Presentation:** 3
**Contribution:** 3
**Rating:** 6
**Confidence:** 3

**Summary:**

The paper proposes a label influence propagation framework for the multi-label node classification task. Specifically, the paper constructs a label influence graph based on the integrated label correlations. Then, the paper propagates high-order influences through this graph and dynamically adjusts the learning process by amplifying labels with positive contributions and mitigating those with negative influence.

**Strengths:**

Pros:
1. The paper considers the positive and negative influence of different labels and encourages or suppresses labels that bring positive or negative influences, respectively.
2. The proposed model is a plug-and-play approach, which can be applied to various GNN backbones.
3. The paper offers a label correlation analysis by dissecting the pipeline into a forward and backward propagation segment.

**Weaknesses:**

Cons:
1. What is the difference between the label propagation methods? (GMNN: Graph Markov Neural Networks, Combining Graph Convolutional Neural Networks and Label Propagation, Resurrecting Label Propagation for Graphs with Heterophily and Label Noise). The paper should cite and compare with them, and highlight the improvement of the model.
2. Some hyperparameters are important to the model. It is better to give some hyperparameter analysis about the model, such as \alpha, \beta. It is suggested to plot the results showing how performance varies with these parameters and report the chosen values.
3. How does the number of label categories k affect the model? It is recommended to study the effect of the performance.
4. It is highly recommended to give some examples, i.e., the visualization of the positive and negative influence of different labels in the case studies, showing how certain labels positively or negatively influence others and how this affects the model's predictions.
5. It is recommended to give other backbones, such as the most commonly used GIN, to show the effectiveness of the model.
6. It is better to check the label of the axis in the figure. i.e., fig 4c. The label of the x-axis is missing.

**Questions:**

Please see the above weakness.

---

> ### Author Response · Authors · 2024-11-21
> **Response to Reviewer jPZg (1/2)**
>
> Dear reviewer jPZg,
> Thank you for your insightful comments. Please, see below our answer to the raised comments/questions.
>
> > **W1. What is the difference between the label propagation methods?**
>
> Thank you very much for your suggestion. We have added a paragraph for label propagation algorithm (LPA) related work in Appendix C. Please refer to it for the complete version of the discussion.
>
> LPA is a classic and insightful algorithm that has inspired a series of related GNN works.
> In GMNN [1], the M-step models label dependencies using a GNN. GCN-LPA [2] not only theoretically analyzed the relationship between GCN and LPA but also incorporated LPA as a regularization term for GCN, improving the performance of single-label node classification.
> R2LP [3] not only generalizes LPA to more realistic scenarios involving heterophily graphs and varying noise levels but also provides a theoretical analysis of its effectiveness on label denoising.
>
> The differences between our Label Influence Propagation (LIP) and the LPA-related works are mainly threefold:
> 1. Propagation target: We do not propagate labels; instead, we propagate the influence between labels.
> 2. Propagation medium: In LPA, the propagation medium is typically the graph structure that connects nodes. In contrast, the propagation medium in LIP is the label propagation graph, constructed by quantified pairwise label influences. In this graph, nodes are labels, and edges are the influence correlations between labels.
> 3. Purpose of propagation: The purpose of propagating label influence is not to infer unknown labels or denoise existing labels, but rather to extend the computed pairwise label influences on high-order label influences. This ultimately encourages labels with positive influence and suppresses those with negative influence.
>
>
> > **W2. How does the performance change when varying the $\alpha, \beta$?**
>
> We have added a hyper-parameter sensitivity study on $\alpha, \beta$ in Appendix E.5.
> The conclusion is that our method demonstrates robustness across a range of hyper-parameter settings. Specifically, the performance remains stable within reasonable parameter ranges, indicating that the model does not heavily depend on fine-tuned hyper-parameters for achieving effective results. Moreover, we find that regardless of the dataset characteristics, the optimal range for $\alpha$ is approximately between 0.08 and 0.30, while the optimal range for $\beta$ is approximately between 0.10 and 0.56. We choose 0.15 and 0.28 respectively for $\alpha, \beta$.
>
> > **W3. How does the number of label categories $k$ affect the model?**
>
> Thank you for this insightful question. Due to time constraints, we only tested the performance of different label quantities on a single dataset, HumLoc. We used GCN as the backbone to compare and analyze the performance of our method across different label number of label categories $k$.
>
> Table 1. Performance (AUC) when changing the number of label categories $k$ on HumLoc.
>
> | HumLoc (AUC) |        2         |        4         |        6         |        8         |        10        |        12        |        14        |
> | :----------: | :--------------: | :--------------: | :--------------: | :--------------: | :--------------: | :--------------: | :--------------: |
> |     GCN      | 70.29 $\pm$ 1.74 | 64.44 $\pm$ 0.92 | 66.35 $\pm$ 1.49 | 65.14 $\pm$ 1.85 | 66.52 $\pm$ 1.22 | 66.73 $\pm$ 0.86 | 68.14 $\pm$ 1.88 |
> |   GCN+LIP    | 73.17 $\pm$ 1.56 | 65.99 $\pm$ 1.02 | 68.41 $\pm$ 1.55 | 68.82 $\pm$ 0.97 | 72.13 $\pm$ 1.29 | 72.22 $\pm$ 1.76 | 75.22 $\pm$ 1.76 |
>
> From the table above, we can observe that our method consistently achieves performance boosting regardless of the number of labels. We also identified some interesting phenomena from the experiments. First, for GCN, the performance is actually the best when the number of labels is 2. This phenomenon can be understood from two perspectives:
> On one hand, statistical analysis shows that label 2 is the category with the highest number of training nodes across all training datasets. Therefore, when the total number of labels is 2, the contribution of label 2 leads to a higher average performance. On the other hand, this also indicates that, to some extent, an increase in the number of labels results in a decline in GCN’s prediction performance.
>
> Moreover, with our method, the performance generally improves as the number of labels increases. In contrast, GCN’s performance fluctuates as the number of labels increases, sometimes improving and sometimes deteriorating. This indirectly indicates that our method reduces the negative influence between labels while enhancing the positive influence.

---

> ### Author Response · Authors · 2024-11-21
> **Response to Reviewer jPZg (2/2)**
>
> > **W4. Give some examples to show the positive and negative influences between labels and how these influences affect the model’s predictions.**
>
> Thank you very much for your suggestion. We have added a case study in Appendix E.8 to better illustrate the positive and negative mutual influences between labels. In this study, we take a specific node and its surrounding neighbors as an example to demonstrate the influences between labels and how these influences contribute to either improving or impairing the prediction of the node.
>
> It is worth noting that, as discussed in Sec. 4, the influence between labels is quite complex. The influence from $Y_a$ to $Y_b$ is the combined effect of a group of nodes with label $y_a$ on a group of nodes with label $y_b$ during both the P and T processes. Here, we attempt to visualize the positive and negative influences, as well as their ultimate effect on performance, using the local structure of a single node rather than a group of nodes with the same label. The case study provides a simplified perspective on these interactions.
>
> > **W5. Try more backbones, such as GIN.**
>
> We have conducted a series of experiments by replacing the backbone with other commonly used GNNs: GIN[4] and APPNP[5]. GIN is a GNN model with theoretical proof of high expressive power. APPNP is a classical yet powerful model.
> We have also added experiments with other alternative backbones in Appendix E.4 of the revised pdf.
> Here, we evaluate the effectiveness of our method, LIP, on these GNNs and present a comparison of the results between the original models and the LIP-enhanced models on MLNC tasks in the table below.
>
> Table 2. Changing backbones (AUC) under node split on MLNC datasets.
>
> | AUC       | DBLP             | BlogCat          | PCG              | EukLoc           |
> |:---------:|:----------------:|:----------------:|:----------------:|:----------------:|
> | GCN       | 92.83 $\pm$ 1.13 | 66.14 $\pm$ 1.74 | 59.54 $\pm$ 0.90 | 70.53 $\pm$ 1.97 |
> | GCN+LIP   | 94.38 $\pm$ 1.51 | 70.21 $\pm$ 2.02 | 67.73 $\pm$ 0.52 | 74.92 $\pm$ 1.82 |
> | GIN       | 93.00 $\pm$ 0.46 | 68.32 $\pm$ 0.67 | 63.44 $\pm$ 1.15 | 73.13 $\pm$ 1.24 |
> | GIN+LIP   | 94.75 $\pm$ 1.29 | 70.87 $\pm$ 0.93 | 66.10 $\pm$ 1.64 | 75.10 $\pm$ 1.29 |
> | APPNP     | 94.17 $\pm$ 0.92 | 70.33 $\pm$ 1.10 | 64.96 $\pm$ 1.33 | 74.67 $\pm$ 0.98 |
> | APPNP+LIP | 95.21 $\pm$ 1.08 | 71.82 $\pm$ 1.45 | 67.51 $\pm$ 1.74 | 75.86 $\pm$ 1.02 |
>
> From the table above, we can observe that even for high-performance GNNs, LIP consistently achieves performance improvements. LIP provides enhancements across different domains and demonstrates exceptional results on certain datasets. For example, LIP improved the performance of GIN by approximately 3% on the PCG dataset.
> In many cases, the performance improvement brought by LIP exceeds the performance differences between different models, highlighting the value of incorporating LIP.
> This is because these backbones primarily focus on modeling the input graph structure and node features. Regardless of the type of GNNs used, our LIP can provide additional support by modeling the influence correlation between labels.
>
>
> > **W6. The label of the x-axis of Fig.4c is missing.**
>
> Thank you for your careful and detailed review. As stated in the original paper on lines 522-523, the horizontal axis of this figure represents the number of epochs. Additionally, we have added have added labels (number of epochs) to the horizontal axis of in Fig. 4c in the revised version.
>
>
> [1] GMNN: Graph Markov Neural Networks.
>
> [2] Combining Graph Convolutional Neural Networks and Label Propagation (GCN-LPA).
>
> [3] Resurrecting Label Propagation for Graphs with Heterophily and Label Noise (R2LP).
>
> [4] How Powerful are Graph Neural Networks? (GIN)
>
> [5] Predict then propagate: Graph neural networks meet personalized pagerank (APPNP).

---

> ### Author Response · Authors · 2024-12-02
>
> Dear reviewer jPZg,
>
> Thank you for the time and effort you have dedicated to reviewing our submission.
>
> We sincerely appreciate your positive rating and hope that our responses to your review comments have addressed your concerns. As the discussion phase is coming to an end, we would like to know if you have any additional questions or suggestions.
>
> Thank you once again.

---

> > ### Comment · Reviewer_jPZg · 2024-12-02
> >
> > I acknowledge the efforts made by the authors and decide to keep my score.

---

> > > ### Author Response · Authors · 2024-12-02
> > >
> > > Dear reviewer jPZg,
> > >
> > > Thank you for your acknowledgment. Your comments and suggestions have made our paper better.

---

### Official Review · Reviewer_d1PZ · 2024-11-07

**Soundness:** 3
**Presentation:** 3
**Contribution:** 3
**Rating:** 8
**Confidence:** 4

**Summary:**

This paper presents Label Influence Propagation (LIP), a novel approach for multi-label node classification (MLNC) on graphs. The key innovation is analyzing and leveraging the mutual influences between different labels, rather than just label correlations. The authors decompose the message passing process into propagation and transformation operations to quantify label influences, construct a label influence graph, and dynamically adjust the learning process based on positive and negative label interactions.

**Strengths:**

1. The paper introduces a new way to analyze label relationships by examining their mutual influences rather than just correlations, supported by empirical observations shown in Figure 1.

2. The work provides a theoretical analysis of how label influences emerge during both propagation and transformation operations in graph neural networks.

3. The proposed LIP framework is plug-and-play compatible with various GNN architectures and shows consistent performance improvements across different datasets and settings.

**Weaknesses:**

The paper doesn't thoroughly discuss how the method scales with increasing numbers of labels or larger graphs.

**Questions:**

1. What are the limitations of decomposing message passing into propagation and transformation operations? Are there cases where this decomposition might not hold?

2. How sensitive is the method to the initial construction of the label influence graph?

---

> ### Author Response · Authors · 2024-11-21
> **Response to Reviewer d1PZ**
>
> Dear reviewer d1PZ,
> Thank you for your insightful comments. Please, see below our answer to the raised comments/questions.
>
> > **W1. How does the model scale with an increasing number of label categories $k$ or nodes $n$?**
>
> Thank you for this insightful question.
> - **Scales with increasing number of label categories $k$.**
>
> Due to time constraints, we only tested the performance of different label quantities on a single dataset, HumLoc.
>
> Table 1. Performance (AUC) when changing the number of label categories $k$ on HumLoc.
>
> | HumLoc (AUC) |        2         |        4         |        6         |        8         |        10        |        12        |        14        |
> | :----------: | :--------------: | :--------------: | :--------------: | :--------------: | :--------------: | :--------------: | :--------------: |
> |     GCN      | 70.29 $\pm$ 1.74 | 64.44 $\pm$ 0.92 | 66.35 $\pm$ 1.49 | 65.14 $\pm$ 1.85 | 66.52 $\pm$ 1.22 | 66.73 $\pm$ 0.86 | 68.14 $\pm$ 1.88 |
> |   GCN+LIP    | 73.17 $\pm$ 1.56 | 65.99 $\pm$ 1.02 | 68.41 $\pm$ 1.55 | 68.82 $\pm$ 0.97 | 72.13 $\pm$ 1.29 | 72.22 $\pm$ 1.76 | 75.22 $\pm$ 1.76 |
>
> From the table above, we can observe that our method consistently achieves performance boosting regardless of the number of labels.
> Moreover, the performance generally improves as the number of labels increases. In contrast, GCN’s performance fluctuates as the number of labels increases, sometimes improving and sometimes deteriorating. This indirectly indicates that our method reduces the negative influence between labels while enhancing the positive influence.
>
> - **Scales with increasing number of nodes $n$.**
>
> In the MLNC task setting, the total number of nodes in the graph is fixed, meaning that the model can access all nodes during training. Therefore, it is not feasible to vary the number of nodes while keeping the number of labels constant. However, as shown in Table 4 of the paper, our method has been evaluated on datasets ranging from graphs with a few thousand nodes to graphs with millions of nodes. The results indicate that our method consistently achieves improvements across these datasets.
> Furthermore, regarding the training cost, theoretical complexity analysis and empirical statistics on runtime and memory consumption are provided in Appendix C.
>
> In conclusion, our method not only achieves satisfactory results but also demonstrates considerable scalability in terms of both $k$ and $n$.
>
>
> > **Q1. What are the limitations of decomposing message passing into the propagation (P) and transformation (T) operations? Are there cases where this decomposition might not hold?**
>
> Thank you for your insightful question. In essence, both P and T are operations inherently present in the message passing process. Some decomposing or decoupling GNN models separate P and T to allow independent adjustment of their layers and positions, enabling deeper models to maintain better performance. However, existing work [1] has shown through extensive experiments that good performance can be achieved regardless of whether P and T are decomposed. Furthermore, improper design of the layer configuration for P and T after decomposing can lead to performance degradation.
>
> Specifically, as discussed in Appendix A.1 of [1], non-decomposing methods such as ResGCN and DenseGCN can still achieve strong performance when the model depth increases (e.g., Figure 4(b) in [1]). *This suggests that decomposing is not the only way to achieve good results.* On the other hand, for some decomposing methods like DAGNN [2], increasing both the P and T layers simultaneously can result in a significant performance drop as the depth increases (e.g., Figure 9(a) in [1]). However, with a slight modification—fixing the number of T layers to 2 while only increasing the P layers—relatively stable performance can be maintained. *This demonstrates that even when using a decomposing approach, specific designs are required to ensure the method’s effectiveness.*
>
> In the context of our work, we focus on analyzing the influence correlations between labels during the P and T operations, rather than designing how P and T should be combined or adjusted to increase model depth. As shown in the backbone replacement experiments in our Appendix E.3, our method achieves performance improvement regardless of whether the backbone is a decomposed or non-decomposed GNNs.
>
> [1] Model Degradation Hinders Deep Graph Neural Networks.
>
> [2] Towards deeper graph neural networks (DAGNN).
>
>
> > **Q2. How sensitive is the method to the initial construction of the label influence graph?**
>
> Actually, our method does not involve any specific initialization for the label influence graph  $G_{LIP}$ . In fact,  $G_{LIP}$  is computed using Equations 4, 9, 10, and 11 from Section 4, directly after the backbone model performs its first predictions.
> In other words, our method does not manually set an initial state or randomly initialize the structure of  $G_{LIP}$.

---

> > ### Comment · Reviewer_d1PZ · 2024-11-26
> >
> > I acknowledge the efforts made by the authors.

---

> > > ### Author Response · Authors · 2024-11-26
> > >
> > > Dear reviewer d1PZ,
> > >
> > > Thank you for taking the time to review our responses and for your acknowledgment. Your comments have made our work better.

---

### Author Response · Authors · 2024-11-21
**General comments to all reviewers**

Dear all reviewers,

We thank you all for dedicating time to provide us with high quality reviews.

Aside from addressing your concerns personally, please note that we have made revisions according to the review comments, with the changes highlighted in blue text for easier identification in the revised PDF.

Specifically, the following updates have been made to the paper pdf:

1. **Expanded Related Work**: Added summaries of Decoupled GNNs and Label Propagation-related work in Sec 2 and Appendix C.

2. **Polished Sec 4.2**: Further refined the method section for better readability and coherence, while correcting typos and minor errors.

3. **Additional Experiments**: Conducted a series of new experiments, including training cost analysis, backbone replacement, more comprehensive ablations, hyperparameter sensitivity analysis, label scalability experiments, inductive setting experiments, and a case study (mainly in Appendix).

4. **Limitations and Future Work**: Discussed the limitations of the work and outlined potential directions for future research.

---

### Meta-Review · Area_Chair_TCbA · 2024-12-20

**Metareview:**

The proposed method models the interactions between labels as both positive and negative influences for the multi-label node classification problem on graphs. It constructs a label influence graph to quantify the relationships between labels and propagates higher-order influences, which contributed to improving classification accuracy. The proposed method demonstrates significant contributions to multi-label node classification on graphs, theoretical analysis, and performance improvements across datasets. While some weaknesses, such as insufficient related work and limited ablation studies, were initially raised by the reviewers, the authors effectively addressed most of these concerns. This paper provides a novel perspective and technique for the multi-label node classification problem, which makes a valuable contribution to the community.

**Additional Comments On Reviewer Discussion:**

During the discussion period, reviewers raised several important points, including insufficient coverage of related work, limited experimental design, lack of detailed reports on computational costs, and the need for improved readability. The authors' responses led all reviewers to acknowledge that most of their concerns had been effectively addressed. In particular, the addition of ablation studies across multiple datasets, hyper-parameter sensitivity analyses, experiments using new backbones, and detailed reporting of training time and GPU memory usage demonstrated the effectiveness of the proposed method and significantly enhanced the overall quality of the paper, which resulted in the increased scores.

---

### Decision · Program_Chairs · 2025-01-22

Accept (Poster)